# CARM1-expressing ovarian cancer depends on the histone methyltransferase EZH2 activity

Sergey Karakashev[1], Hengrui Zhu[1], Shuai Wu[1], Yuhki Yokoyama[1], Benjamin G. Bitler[1], Pyoung-Hwa Park[1], Jeong-Heon Lee[2], Andrew V. Kossenkov[3], Krutika Satish Gaonkar[4], Huihuang Yan[4], Ronny Drapkin [5], Jose R. Conejo-Garcia[6], David W. Speicher[7], Tamas Ordog [2] & Rugang Zhang [1]

CARM1 is an arginine methyltransferase that asymmetrically dimethylates protein substrates on arginine residues. CARM1 is often overexpressed in human cancers. However, clinically applicable cancer therapeutic strategies based on CARM1 expression remain to be explored. Here, we report that EZH2 inhibition is effective in CARM1-expressing epithelial ovarian cancer. Inhibition of EZH2 activity using a clinically applicable small molecule inhibitor significantly suppresses the growth of CARM1-expressing, but not CARM1-deficient, ovarian tumors in two xenograft models and improves the survival of mice bearing CARM1-expressing ovarian tumors. The observed selectivity correlates with reactivation of EZH2 target tumor suppressor genes in a CARM1-dependent manner. Mechanistically, CARM1 promotes EZH2-mediated silencing of EZH2/BAF155 target tumor suppressor genes by methylating BAF155, which leads to the displacement of BAF155 by EZH2. Together, these results indicate that pharmacological inhibition of EZH2 represents a novel therapeutic strategy for CARM1-expressing cancers.

[1] Gene Expression and Regulation Program, The Wistar Institute, Philadelphia, PA 19104, USA. [2] Epigenomics Program, Center for Individualized Medicine, Mayo Clinic, Rochester, MN 55905, USA. [3] Center for Systems and Computational Biology, The Wistar Institute, Philadelphia, PA 19104, USA. [4] Division of Biostatistics and Informatics, Department of Health Science Research, Mayo Clinic, Rochester, MN 55905, USA. [5] Department of Obstetrics and Gynecology, Perelman School of Medicine, University of Pennsylvania, Philadelphia, PA 19104, USA. [6] Department of Immunology, Moffitt Cancer Center, Tampa, FL 33612, USA. [7] Molecular and Cellular Oncology Program, The Wistar Institute, Philadelphia, PA 19104, USA. Correspondence and requests for materials should be addressed to R.Z. (email: rzhang@wistar.org)

CARM1, also known as PRMT4, is a type I protein arginine methyltransferase (PRMT) that asymmetrically dimethylates protein substrates on arginine residues[1]. CARM1 is located at 19p13.2[1]. Emerging evidence suggests CARM1 functions as an oncogene in human cancers[1]. High levels of CARM1 expression have been observed in several cancer types, including breast, colon, and prostate[2–4]. CARM1 stimulates cell growth in breast cancer[5,6]. CARM1 knockout mice die at birth, indicating that CARM1 is specifically required for postnatal survival[7]. Knockin of methyltransferase-inactivated CARM1 phenocopies CARM1 null mice, demonstrating that CARM1's enzymatic activity is required for postnatal survival[8]. Although small molecule inhibitors of CARM1 have been reported based on biochemical screening[9,10], there is no evidence that they can be administered without toxicity in vivo. It is possible that targeting CARM1 activity is impossible given that it is required for postnatal survival[7]. Thus, despite its oncogenic role, clinically applicable therapeutic strategies based on CARM1 expression in cancer remain to be explored.

The development of novel therapeutic strategies for ovarian cancers remains a major obstacle to overcome. Epithelial ovarian cancer (EOC) remains the most lethal gynecological malignancy in the United States[11]. Recent discoveries have demonstrated that EOC is composed of multiple separate diseases[12]. High-grade serous ovarian cancer (HGSOC) is the most common subtype (>70% of EOC cases) and accounts for the majority of EOC-associated mortalities[12]. EOC is genetically heterogeneous[12]. Thus, it is imperative that therapeutic strategies need to be personalized by targeting distinct molecular subsets of EOC[13]. Notably, the role of CARM1 in EOC has not been explored.

CARM1 has been shown to methylate substrates involved in epigenetic chromatin remodeling[1]. This suggests that epigenetic mechanisms play a key role in CARM1-expressing cancers. EZH2 is the catalytic subunit of the polycomb repressive complex 2 (PRC2), which silences its target genes by generating the lysine 27 trimethylation epigenetic mark on histone H3 (H3K27Me3)[14]. EZH2 is overexpressed in EOC[15,16]. Notably, EZH2 inhibitors are safe in clinical trials for hematopoietic malignancies[17].

Here, we show that inhibition of EZH2 activity is selective against CARM1-expressing EOC. Specifically, inhibition of EZH2 methyltransferase activity by clinically applicable small molecule inhibitors such as GSK126 suppresses the growth of CARM1-high, but not CARM1-low, HGSOC in both orthotopic and patient-derived xenograft (PDX) mouse models. This correlates with an improvement of survival of mice bearing CARM1-high HGSOC. Mechanistically, CARM1 promotes EZH2-mediated silencing of target tumor suppressor genes. This correlates with the displacement of BAF155, a subunit of the SWI/SNF chromatin remodeling complex[18], by EZH2 through methylation of BAF155 by CARM1. Thus, our findings provide scientific rationale for targeting CARM1 expression in EOC using pharmacological inhibition of EZH2 activity.

## Results

**EZH2 inhibitors are selective against CARM1-high cells.** Analysis of high-throughput genetic profiles from The Cancer Genomics Atlas (TCGA) revealed amplification of CARM1 in ~10% of HGSOC (Supplementary Fig. 1a)[19], which correlates with a significantly higher level of CARM1 expression (Supplementary Fig. 1b). Consistently, CARM1 was expressed at a higher level in laser captured and microdissected (LCM) HGSOCs compared with normal human ovarian surface epithelial (HOSE) cells (Supplementary Fig. 1c)[20]. Recent evidence indicates that the majority of HGSOC likely develop from the fallopian tube fimbriae epithelium (FTE)[13,21]. Indeed, CARM1 was also expressed

at a higher level in LCM HGSOCs compared with normal human FTE cells (Supplementary Fig. 1d)[22]. Likewise, CARM1 was expressed at higher levels in a number of EOC cell lines compared with either FTE or HOSE cells (Fig. 1a). Interestingly, CARM1 amplification and BRCA1/2 mutations do not typically occur in the same tumor (Supplementary Fig. 1a)[19]. CARM1 amplification predicted a shorter overall survival in TCGA HGSOC database (Supplementary Fig. 1e). Thus, CARM1 is amplified in EOC, and its amplification/high expression correlates with a poor overall survival in EOC patients.

Toward understanding the role of CARM1 in EOC, we generated a CARM1 knockout (CARM1 KO) clone in CARM1-high A1847 cells using the CRISPR methodology (Fig. 1b). Consistent with CARM1's growth-promoting role reported in other cancer types[1], CARM1 KO cells exhibited a decrease in growth compared with parental controls (Supplementary Fig. 1f). Similar observations were made with shRNA-mediated CARM1 knockdown in CARM1-high EOC cell lines such as OVCAR10 and A1847 (Supplementary Fig. 1g–j). Thus, CARM1 inhibition suppresses the growth of EOC cells.

CARM1 asymmetrically dimethylates substrates involved in epigenetic regulation of gene transcription[1]. This suggests that epigenetic mechanisms play a key role in mediating the oncogenic activity of CARM1. Thus, we performed an unbiased evaluation of a set of 23 small molecule epigenetic inhibitors[23]. We evaluated each individual inhibitor for its ability to selectively suppress the growth of CARM1-expressing cells compared with CARM1 KO cells. Interestingly, both of the EZH2 inhibitors in the set (namely, GSK126 and UNC1999) showed selectivity against CARM1-expressing cells (Fig. 1c, Supplementary Fig. 1k and Supplementary Table 1)[24,25]. This is not due to a reduced proliferation of CARM1 KO cells because (1) a number of small molecule inhibitors were equally effective in suppressing the growth of both CARM1-expressing and KO cells (e.g., CI994, Figs. 1d, 2); and (2) we normalized the data to the growth of vehicle-treated CARM1-expressing or KO cells to control for variation in cell growth. The observed CARM1-dependent selectivity by EZH2 inhibitors was not due to changes in EZH2 levels because its expression was not altered and levels of its enzymatic product H3K27Me3 were not changed by CARM1 knockout (Fig. 1e). EZH2 is overexpressed HGSOCs and EOC cell lines compared to either HOSE cells or FTE cells regardless of CARM1 levels (Fig. 1a and Supplementary Fig. 2a, b)[19,22]. However, CARM1 does not affect EZH2 expression (Fig. 1e). Consistently, there was no significant difference in EZH2 levels between HGSOCs with or without CARM1 amplification or overexpression in the TCGA database (Fig. 1f and Supplementary Fig. 2c).

Since GSK126 is currently in clinical development[17] and UNC1999 is less selective than GSK126[25], we focused our studies on GSK126. We determined that 10 μM GSK126 was sufficient to inhibit >95% of the enzymatic activity of EZH2 as indicated by the decrease in H3K27Me3 levels (Supplementary Fig. 2d). Note that EZH2 inhibition did not affect the expression of CARM1. Consistent with previous reports[23,24], GSK126 did not affect EZH2 protein levels but instead selectively inhibited its methyltransferase activity as evidenced by a dose-dependent decrease in H3K27Me3 levels. Thus, we used 10 μM GSK126 for subsequent studies. Validating our pharmacological screen, there was a correlation between CARM1 expression levels and cellular response to GSK126 in a panel of EOC cell lines (Figs. 1a, 2a, b and Supplementary Fig. 2e). Similar observations were also made in primary cultures of CARM1-expressing HGSOCs treated with GSK126 (Supplementary Fig. 2f–h). Notably, GSK126 did not affect the growth of either HOSE or FTE cells (Fig. 2a). We obtained similar results in CARM1-high parental and CARM1

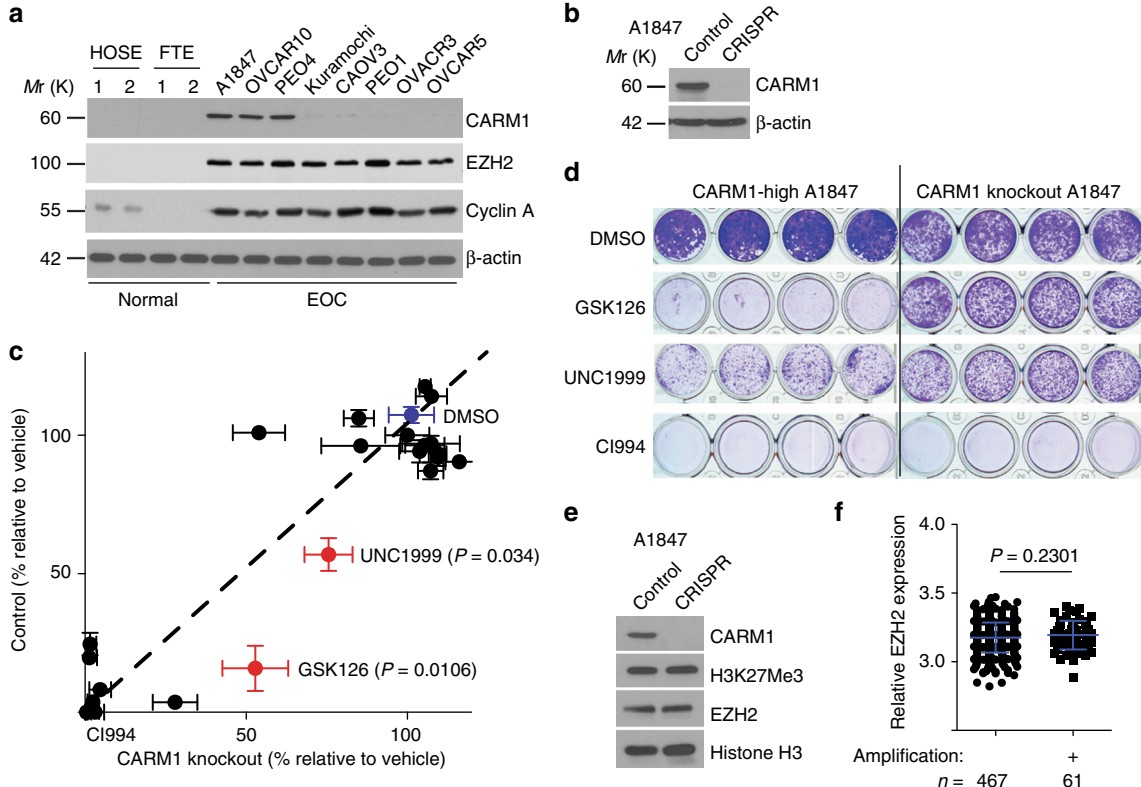

**Fig. 1** CARM1-expressing EOC cells are selectively sensitive to EZH2 inhibitors. **a** Expression of CARM1, EZH2, and Cyclin A in the indicated EOC cell lines, HOSE, and FTE cells were determined by immunoblot. Expression of β-actin was used as a loading control. **b** Expression of CARM1 and a loading control β-actin in CARM1-high parental and CRISPR-mediated CARM1 knockout (KO) A1847 cells. **c** Equal number of parental control or CARM1 knockout A1847 cells were plated and treated with each of the 23 individual epigenetic inhibitors for 14 days. The media with the inhibitors was refreshed every 3 days. Cell growth was quantified as integrated density using NIH ImageJ software. Quantification of the average integrated density graphed as a scatter plot. The *X*-axis indicates the relative growth of treated CARM1 knockout A1847 cells compared with DMSO vehicle controls. *Y*-axis indicates the relative growth of treated CARM1-high parental A1847 cells compared with DMSO vehicle controls. *n* = 4; error bars represent SEM. **d** Representative images of colonies formed by the cells treated with the indicated inhibitors. GSK126 and UNC1999 represent positive hits from the screen. Note that CI994 was used a negative control that showed no difference between parental and CARM1 knockout cells. **e** Expression of CARM1, H3K27Me3, EZH2, and H3 in parental control and CARM1 knockout A1847 cells. **f** Relative expression of EZH2 in the TCGA HGSOC cases with (*n* = 61) or without (*n* = 467) *CARM1* amplification. Relative EZH2 expression levels were transformed to log$_2$ (10+ expression) values. *P*-values are from two-tailed *t*-test. Error bars represent SD

KO cells in 3D cultures (Supplementary Fig. 2i–j). Notably, ectopic CARM1 expression sensitized CARM-low EOC cells to GSK126 (Fig. 2c, d and Supplementary Fig. 2k–l). EZH2 is expressed at comparable levels in CARM1-high EOC cells and CARM1-low EOC cells that did not respond to GSK126 such as CAOV3 and PEO1 (Figs. 1a, 2a). In addition, markers of cell proliferation such as Cyclin A were expressed at comparable levels in CARM1-high and CARM1-low EOC cells (Figs. 1a, 2a). Consistent with the observed selectivity, markers of apoptosis were induced by EZH2 inhibition in a CARM1-dependent manner (e.g., Fig. 2e, f and Supplementary Fig. 2m). These data point to sensitivity to EZH2 inhibitors as a unique and exploitable therapeutic vulnerability in CARM1-high EOCs.

To limit potential off-target effects and validate that the observed effects were due to inhibition of EZH2's methyltransferase activity, we performed genetic rescue experiments. Indeed, apoptosis induced by EZH2 knockdown could be rescued by wild-type EZH2 but not by a mutant with inactivated catalytic activity (Fig. 2g, h). Consistently, the cell growth inhibition induced by EZH2 knockdown was rescued by wild-type EZH2 but not by a catalytically inactive EZH2 mutant (Fig. 2i, j). Recent evidence suggests that EZH2 inhibitor can also affect cell growth by destabilizing PRC2 complex in a catalytic activity-independent

manner[26]. GSK126 did not weaken the interaction between PRC2 subunits EZH2 and SUZ12 (Supplementary Fig. 2n), indicating that the observed selectivity against CARM1 was not due to destabilization of PRC2 complex. Together, we conclude that the observed selectivity against CARM1 by EZH2 inhibitor is due to the inhibition of its methyltransferase activity.

**CARM1 promotes the silencing of EZH2 target genes**. To explore the mechanistic basis of the selectivity against EZH2 inhibitor, we performed RNA-deep sequencing (RNA-Seq) in parental control and CARM1 KO cells (Fig. 3a). To identify direct EZH2 target genes that are regulated by CARM1, we performed EZH2 and H3K27Me3 chromatin immunoprecipitation followed by deep sequencing (ChIP-seq) analysis in control and CARM1 KO cells (Fig. 3a). Since CARM1 may promote EZH2-dependent gene silencing, we focused on the genomic loci that showed a decrease in association with EZH2/H3K27Me3 in CARM1 KO cells compared with controls. Cross-referencing RNA-Seq and ChIP-seq data (GEO access number: GSE95645) revealed a list of 218 direct EZH2/H3K27Me3 target genes that were down-regulated by CARM1 (>3-fold) (Fig. 3a and Supplementary Fig. 3a, b). This represents a 8.3-fold enrichment of EZH2/

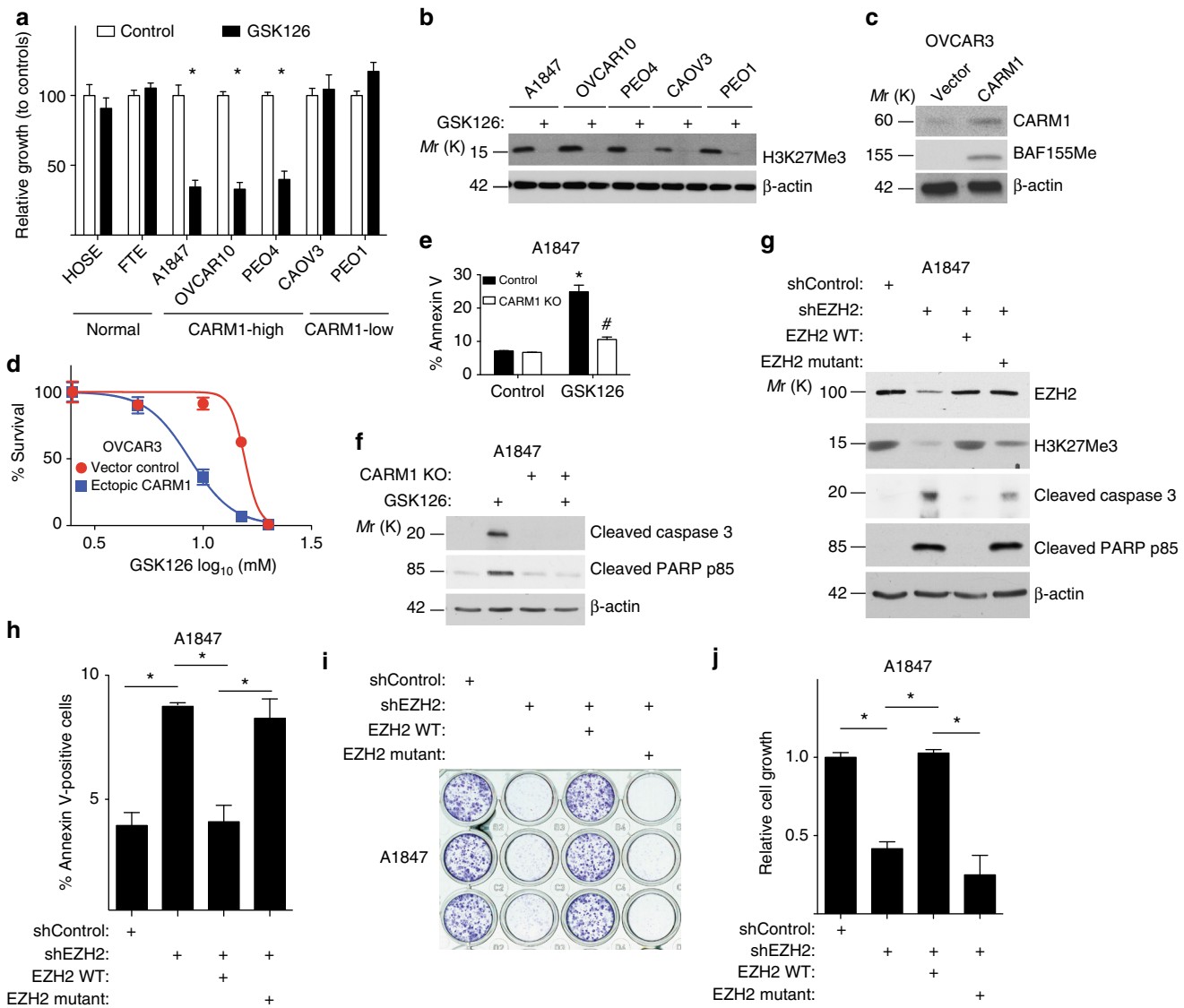

**Fig. 2** The selectivity against CARM1 by EZH2 inhibition correlates with apoptosis induction in a methyltransferase activity-dependent manner. **a** Relative growth of the indicated HOSE, FTE, and EOC cancer cell lines with high or low CARM1 expression treated with 10 μM GSK126 or vehicle in a colony formation assay as determined by NIH ImageJ quantification. **b** Expression of H3K27Me3 in the indicated EOC cell lines with high or low CARM1 expression treated with or without 10 μM GSK126. Expression of β-actin was used as a loading control. **c, d** Expression of CARM1, BAF155Me, and β-actin in OVCAR3 cells with ectopic CARM1 or control (**c**). GSK126 dose response curves of the indicated cells were determined by colony formation assay (**d**). Mean of three independent experiments with SEM. **e** Control parental and CARM1 knockout A1847 cells were treated with 10 μM GSK126 or vehicle DMSO control for 7 days. Percentage of Annexin V-positive apoptotic cells was quantified. *$P < 0.001$ and #$P > 0.05$. **f** Apoptosis markers cleaved caspase 3 and cleaved PARP p85 in parental and CARM1 knockout A1847 cells treated with 10 μM GSK126 for 7 days. Expression of β-actin was used as a loading control. **g–j** CARM1-high A1847 cells were infected with a lentivirus encoding shEZH2 targeting the 3′ untranslated region (UTR) of the human *EZH2* gene together with a retrovirus encoding wild-type EZH2 (WT) or a SET domain-deleted EZH2 mutant (EZH2 ΔSET). Drug-selected cells were examined for EZH2, H3K27Me3, apoptosis markers cleaved caspase 3 and cleaved PARP p85, and a loading control (β-actin) by immunoblot (**g**), quantified for Annexin V-positive apoptotic cells by FACS (**h**), subjected to a colony formation assay (**i**), and quantified for the relative cell growth based on colony formation using NIH ImageJ (**j**). *$P < 0.01$. Means of three independent experiments with SEM. *P*-values are from two-tailed *t*-test

H3K27Me3 target genes among the 2084 genes upregulated at least 3-fold in CARM1 KO cells compared with parental controls ($P = 1.5 \times 10^{-136}$, determined by hypergeometric test) (Supplementary Fig. 3c). We cross-referenced the list of 218 direct EZH2/H3K27Me3 target genes with 528 TCGA high-grade serous ovarian carcinomas gene expression profiles and identified genes that negatively correlated with CARM1 expression in these cases. These prioritizations led to a list of 36 EZH2 direct target genes that negatively correlated with CARM1 expression. Pathway analysis revealed that the top functional pathway enriched in

these genes was apoptosis ($n = 19$, $P = 2.6 \times 10^{-6}$, determined by Fisher's Exact Test) (Fig. 3a and Supplementary Table 2). Notably, three of the ranked apoptosis-regulating genes (*DAB2*[27], *DLC1*[28,29], and *NOXA*[30]) are known tumor suppressors that are implicated in suppressing proliferation and promoting apoptosis (Fig. 3b), i.e., the phenotypes we observed when CARM1 was knocked out or EZH2 activity was inhibited with GSK126 (Figs. 1, 2).

In parallel, we performed Ingenuity Pathway Analysis for the upstream transcription factors that regulate the genes

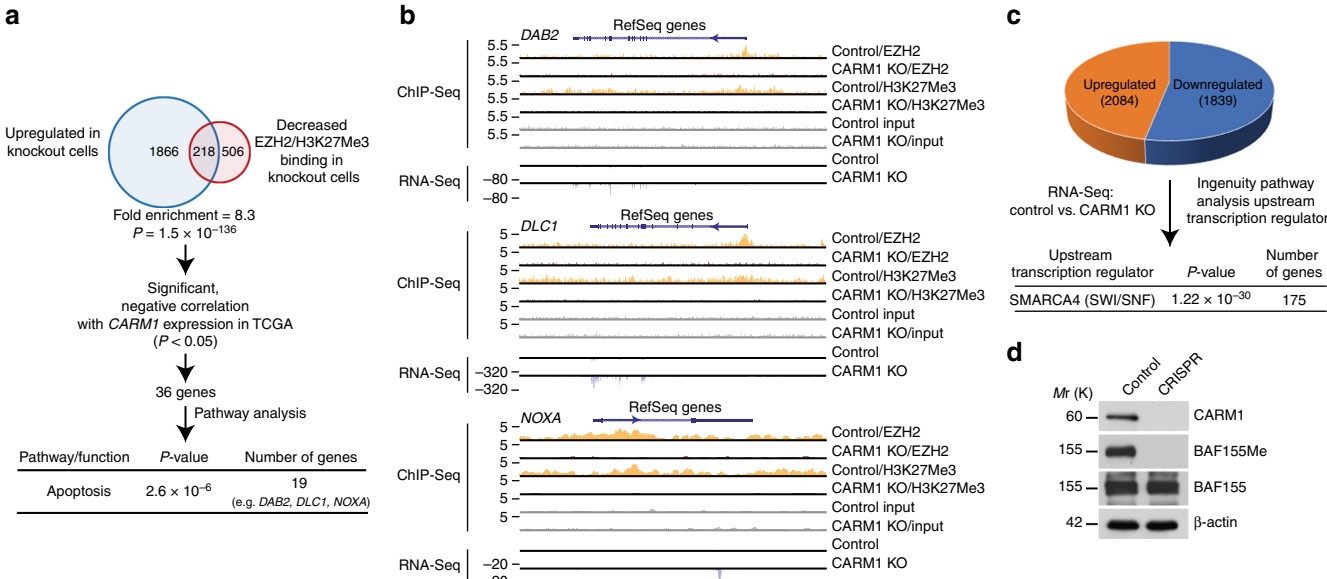

**Fig. 3** CARM1 promotes the silencing of EZH2 target genes and CARM1-regulated genes are enriched for SWI/SNF target genes. **a** Experimental strategy used to identify CARM1-regulated, clinically relevant EZH2/H3K27Me3 target genes. Ingenuity Pathway Analysis revealed that the top pathway enriched for the identified CARM1-regulated EZH2/H3K27Me3 target genes was apoptosis. Specifically, 19 of the 36 identified genes (including *DAB2*, *DLC1*, and *NOXA*) are known to promote apoptosis (see full list in Supplementary Table 2). **b** Examples of EZH2 and H3K27Me3 ChIP-seq and RNA-Seq tracks of the newly identified CARM1-regulated EZH2/H3K27Me3 target genes in the indicated parental control and CARM1 knockout A1847 cells. **c** Differentially expressed genes in control parental and CARM1 knockout A1847 cells identified by RNA-Seq (>3-fold) were subjected to Ingenuity pathway analysis for upstream regulators. The analysis revealed that SMARCA4 (also known as BRG1, a catalytic subunit of the SWI/SNF complex) was the top upstream regulator of these differentially expressed genes. **d** Expression of CARM1, BAF155, and BAF155Me in control parental and CARM1 knockout A1847 cells. Expression of β-actin was used as a loading control

differentially expressed in parental control and CARM1 KO cells. The top upstream transcription regulator identified was SMARCA4 (also known as BRG1), the catalytic subunit of the SWI/SNF complex[31] ($P = 1.22 \times 10^{-30}$, determined by Fisher's Exact Test) (Fig. 3c and Supplementary Table 3). Inhibition of EZH2 activity is synthetically lethal with inactivation of the SWI/SNF complex due to the antagonistic regulation of the same set of genes by the EZH2/PRC2 and the SWI/SNF complexes[18]. CARM1 methylates the R1064 residue of BAF155, a core subunit of the SWI/SNF complex[6]. Indeed, R1064 methylated BAF155 (BAF155Me) levels correlated with CARM1 expression in the panel of the tested cell lines (Fig. 1a and Supplementary Fig. 4a) and the observed changes in selectivity against CARM1 KO or knockdown cells correlated with the decreased BAF155Me levels (Fig. 3d and Supplementary Fig. 4b). In addition, ectopic CARM1 expression in CARM1-low EOC cells correlated with an increase in BAF155Me levels (Fig. 2c and Supplementary Fig. 2k). However, EZH2 inhibition did not affect BAF155Me levels (Supplementary Fig. 4c). Together, these data suggest that CARM1 may promote EZH2-mediated silencing by altering the antagonism between PRC2 and SWI/SNF complex via BAF155 methylation.

**CARM1 regulates antagonism between BAF155 and EZH2.** The expression of the identified genes was upregulated by either EZH2 inhibition by GSK126 treatment or CARM1 knockout or knockdown in CARM1-high cells (Fig. 4a and Supplementary Fig. 4d). GSK126 treatment and CARM1 knockout did not have additive effects on the expression of these genes (Fig. 4a), indicating that they probably function in the same pathway. As a control, known CARM1-regulated BAF155Me target genes such as the tumor suppressor *TIMP3*[6] were downregulated by CARM1

KO but not by EZH2 inhibitor GSK126 (Supplementary Fig. 4e). This suggests that CARM1 promotes the silencing of EZH2/BAF155 target genes in an EZH2-dependent manner, but mediates the expression of CARM1-regulated BAF155Me target genes in an EZH2-independent manner.

ChIP analysis validated that the association of EZH2 and its enzymatic product H3K27Me3 with these gene loci was indeed CARM1-dependent (Fig. 4b, c and Supplementary Fig. 4f). Consistent with previous reports[23,24], EZH2 inhibitor decreased H3K27Me3 occupancy without affecting EZH2's association with its target genes (Fig. 4b, c). Importantly, CARM1 KO led to loss of EZH2 from these target gene loci and a corresponding increase in the association of BAF155 with these gene loci (Fig. 4d and Supplementary Fig. 4g). Conversely, ectopic CARM1 expression in CARM1-low EOC cells decreased BAF155's association with these genes (Fig. 4e). This result indicates that there is a switch from EZH2 to unmethylated BAF155 in these gene loci when CARM1 is knocked out. Consistently, in CARM1-low cells, BAF155 knockdown increased EZH2's association with these genes (Fig. 4f), which correlates with a decrease in their expression (Supplementary Fig. 4h). Notably, BAF155's association with these gene loci correlated with changes in other SWI/SNF components such as BRG1 and SNF5 (Fig. 4g, h). This supports the notion that the observed changes are SWI/SNF complex-dependent. Finally, the association of RNA polymerase II (Pol II) with the gene loci correlated with changes in their expression (Fig. 4a, i). In contrast, there was no significant enrichment of either EZH2 or H3K27Me3 in the promoter of the CARM1-regulated BAF155Me target genes such as *TIMP3*[6] (Supplementary Fig. 4i–j). CARM1 KO but not EZH2 inhibitor GSK126 treatment decreased the association of BAF155, SNF5, BRG1, and Pol II with the *TIMP3* promoter (Supplementary Fig. 4k–n). Our data support a model that CARM1 promotes the

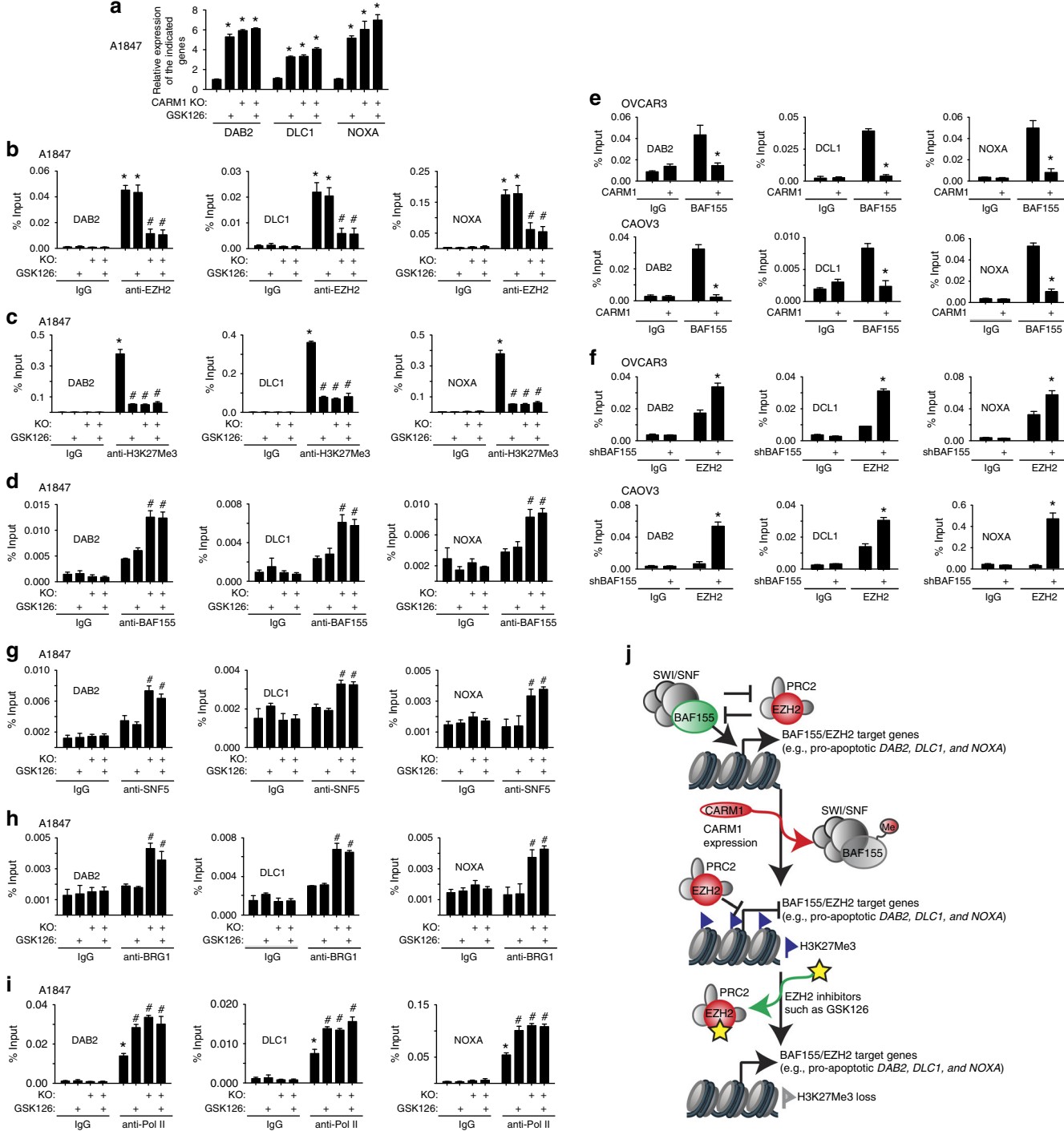

**Fig. 4** CARM1 regulates the antagonism between EZH2 and BAF155 at the promoters of EZH2/BAF155 target genes. **a** Parental control and CARM1 knockout A1847 cells were treated with 10 μM GSK126 or vehicle controls. mRNA expression of the indicated genes was determined by qRT-PCR. *$P < 0.002$ compared with parental controls. **b–d** Parental control and CARM1 knockout A1847 cells were treated with 10 μM GSK126 or vehicle DMSO controls. The cells were subjected to ChIP analysis using anti-EZH2 (**b**), anti-H3K27Me3 (**c**), or anti-BAF155 (**d**) antibodies. *$P < 0.001$ compared with IgG controls and #$P < 0.001$ compared with parental controls. **e** The indicated CARM1-low cells with or without ectopic CARM1 expression were subjected to ChIP analysis using an anti-EZH2 antibody or an isotype-matched IgG control. *$P < 0.05$ compared with vector controls. **f** The indicated CARM1-low cells with or without shBAF155 expression were subjected to ChIP analysis using an anti-BAF155 antibody or an isotype-matched IgG control. *$P < 0.03$ compared with shControls. **g–i** Parental control and CARM1 knockout A1847 cells were treated with 10 μM GSK126 or vehicle DMSO controls. The cells were subjected to ChIP analysis using anti-SNF5 (**g**), anti-BRG1 (**h**), or anti-RNA Pol II (**i**) antibodies. ChIP products were subjected to qPCR analysis using primers specific for the promoter regions of the human *DAB2*, *DLC1*, and *NOXA* genes. *$P < 0.001$ compared with IgG controls and #$P < 0.001$ compared with parental controls. **j** Model proposed for the molecular basis of the selectivity against CARM1 expression by EZH2 inhibition. Means of three independent experiments with SEM. *P*-values are from two-tailed *t*-test

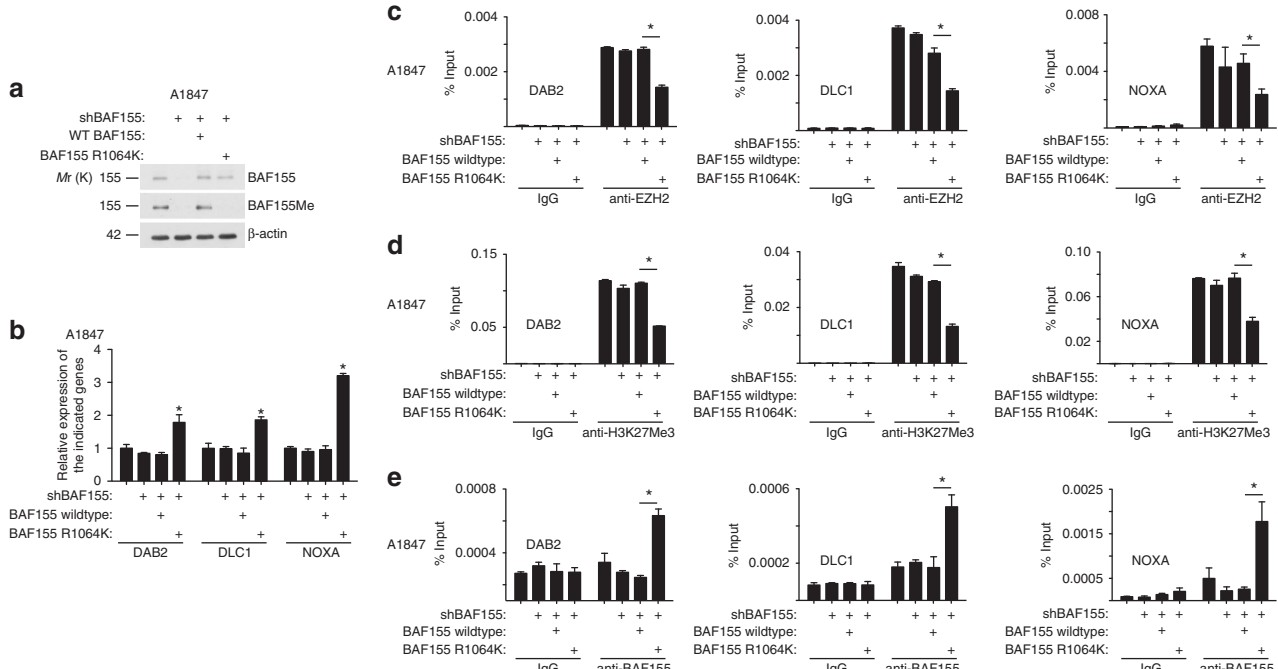

**Fig. 5** CARM1-mediated R1064 BAF155 methylation displaces BAF155 with EZH2 at the promoters of EZH2/BAF155 target genes. **a** CARM1-expressing A1847 cells were infected with a lentivirus encoding shBAF155 targeting the 3′ untranslated region (UTR) of the human *BAF155* gene together with a retrovirus encoding wild-type BAF155 (WT) or a BAF155 R1064K mutant. Expression of BAF155 and BAF155Me was determined by immunoblot. β-actin expression was used as a loading control. **b** Same as **a**, but examined for the expression of the indicated BAF155/EZH2 target genes by qRT-PCR. *$P <$ 0.001. **c**–**e** The indicated cells were subjected to ChIP analysis using anti-EZH2 (**c**), anti-H3K27Me3 (**d**), or anti-BAF155 (**e**) antibodies. An isotype-matched IgG was used as a negative control. ChIP products were subjected to qPCR analysis using primers specific for the promoter regions of the human *DAB2*, *DLC1*, and *NOXA* genes. Error bars represent SEM. *P*-values are from two-tailed *t*-test

silencing of EZH2/BAF155 target genes by displacing BAF155 via methylation, which then permits the occupancy of the target gene promoters by EZH2 and their consequent repression by H3K27Me3. In contrast, CARM1 mediates the expression of CARM1-regulated BAF155Me targeted genes in an EZH2-independent manner. Thus, CARM1 regulates the antagonism between SWI/SNF and PRC2 through methylating BAF155 (Fig. 4j).

We next sought to directly test the effects of CARM1-mediated BAF155 methylation at the R1064 residue on the expression of identified EZH2/BAF155 target genes to establish that the observed antagonism is BAF155Me-dependent. Toward this goal, in CARM1-high A1847 cells, we replaced endogenous BAF155 with either mutant BAF155 that can no longer be methylated by CARM1 (BAF155R1064K) or wild-type BAF155 (Fig. 5a). Indeed, BAF155 R1064K mutant but not wild-type BAF155 upregulated the expression of the EZH2/BAF155 target genes (Fig. 5b), indicating that only the unmethylated BAF155 can be associated with these genes. This correlated with a decrease in EZH2 and its enzymatic product H3K27Me3 at the promoter of these genes and a concurrent increase of BAF155's association with these gene promoters (Fig. 5c, d). In contrast, the association of BAF155 with the BAF155Me target gene *TIMP3* was rescued by wild-type BAF155 but not the BAF155 R1064K, which correlated with the suppression of *TIMP3* by BAF155 R1046K but not wild-type BAF155 (Supplementary Fig. 5a, b). Together, we conclude that CARM1-mediated methylation of BAF155 drives a switch from BAF155 to EZH2 at the promoters of the BAF155/EZH2 target tumor suppressor genes (Fig. 4f). Therefore, EZH2 inhibition reactivates the tumor suppressive BAF155/EZH2 target genes to promote apoptosis and inhibit proliferation of CARM1-expressing cells (Fig. 4f).

**GSK126 improves the survival of CARM1-high tumor bearing mice**. EZH2 inhibitors such as GSK126 are safe in clinical trials for hematopoietic malignancies[17]. To determine the effects of EZH2 inhibition in vivo on the growth of CARM1-expressing ovarian tumors, we utilized two xenograft models. In the subcutaneous xenograft models, the injected CARM1-expressing A1847 cells were first allowed to grow for 1 week to establish the tumors. Mice were then randomized and treated daily with vehicle control or GSK126 (50 mg/kg) by intraperitoneal (i.p.) injection[23,24]. Indeed, GSK126 treatment significantly inhibited the growth of CARM1-expressing tumors (Supplementary Fig. 6a, b). In contrast, GSK126 failed to inhibit the growth of tumors formed by CARM1 knockout A1847 cells (Supplementary Fig. 6c). Ectopic CARM1 expression sensitized CARM1-low OVCAR3 tumors to GSK126, while GSK126 did not significantly affect the growth of CARM1-low OVCAR3 tumors (Supplementary Fig. 6d). Notably, ectopic CARM1 promoted the growth of tumors formed by CARM1-low OVCAR3 cells (Supplementary Fig. 6d). Consistently, GSK126 significantly suppressed the growth of CARM1-high, but not CARM1-low, high-grade serous PDXs (Supplementary Fig. 6e–g). To more closely mimic the tumor microenvironment, we orthotopically transplanted A1847 cells into the bursa covering the ovary of immunocompromised NSG mice. Similarly, the injected cells were first allowed to grow for 1 week to establish the tumors. Mice were then randomized and treated daily with vehicle control or GSK126 (50 mg/kg) by i. p. injection. Similar to subcutaneous xenograft models, the growth of CARM1-expressing tumors was significantly inhibited by GSK126 in the orthotopic xenograft models (Fig. 6a, b). We next followed the survival of the treated mice after stopping the treatment. Importantly, GSK126 significantly improved the survival of mice bearing the orthotopically transplanted CARM1-

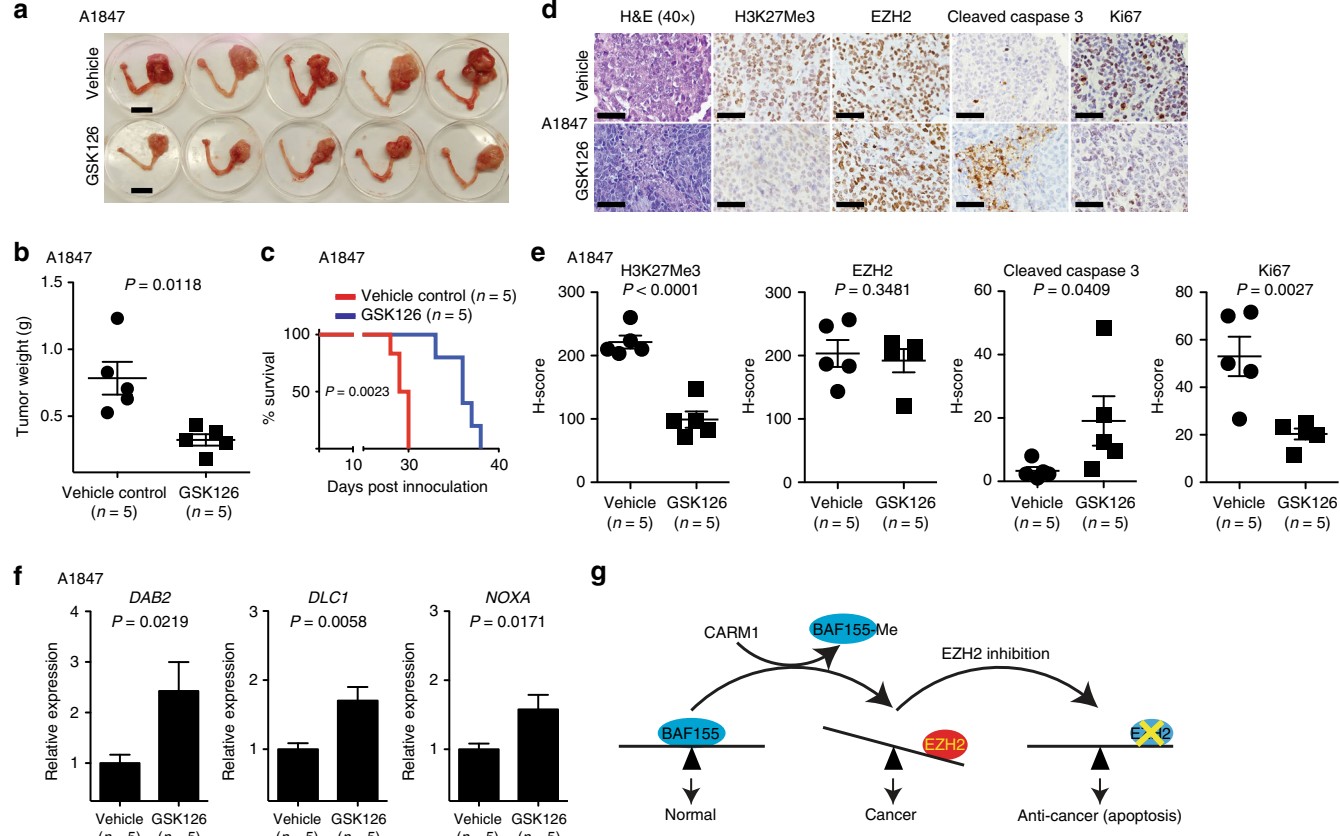

**Fig. 6** EZH2 inhibition suppressed the growth of CARM1-expressing tumors in vivo and improved the survival of mice bearing CARM1-expressing tumors. **a** CARM1-high A1847 ovarian cancer cells were unilaterally injected into the ovarian bursa sac of immunocompromised mice (n = 5/group). Tumors were allowed to establish for 1 week before the mice were randomized into two different treatment groups. Mice were treated with vehicle control or GSK126 (50 mg/kg daily) for an additional 3 weeks. At the end of treatment, the mice were euthanized. Shown are images of reproductive tracts with tumors from control or GSK126-treated mice. Bar = 1 cm. **b** Tumor weight was measured as a surrogate for tumor burden from the control and GSK126-treated mice. **c** After stopping the treatment, the mice from the indicated groups were followed for survival. Shown is the Kaplan–Meier survival curve for GSK126 or vehicle-treated mice. P-value was calculated by log-rank test. **d** The serial sections of tumors dissected from the indicated treatment groups were subjected to immunohistochemical staining for H3K27Me3, EZH2, cleaved caspase 3, and Ki67. Scale bar = 50 μm. **e** Histological score (H-score) of the indicated proteins was calculated for three separate fields from five tumors from five individual mice from each of the indicated groups. **f** Expression of the indicated EZH2/BAF155 target genes was determined by qRT-PCR in the tumors dissected from the indicated treatment groups. **g** CARM1 regulates the antagonism between SWI/SNF and PRC2 through methylating BAF155. EZH2 inhibition reactivates EZH2/BAF155 target tumor suppressive genes to promote apoptosis, inhibit proliferation, and suppress tumor growth. Error bars represent SEM. P-values are from two-tailed t-test

expressing tumors compared to controls (Fig. 6c) (P = 0.0023, determined by log-rank test). We conclude that the EZH2 inhibitor GSK126 significantly suppressed the growth of CARM1-expressing tumors and improved the survival of mice bearing these tumors.

We performed immunohistochemical (IHC) analysis for markers of cell proliferation (Ki67), apoptosis (cleaved caspase 3), H3K27Me3, and EZH2. H3K27Me3 staining was decreased by GSK126, while GSK126 did not affect EZH2 staining (Fig. 6d, e). Further, GSK126 treatment decreased the expression of Ki67 and increased the expression of cleaved caspase 3 (Fig. 6d, e). Finally, the observed decrease in cell proliferation and increase in apoptosis correlated with the upregulation of the identified CARM1-regulated EZH2/BAF155 target genes such as *DAB2*, *DLC1*, and *NOXA* by the EZH2 inhibitor GSK126 in vivo (Fig. 6f). In contrast, GSK126 did not affect the expression of Ki67 and cleaved caspase 3 in tumors formed by CARM1 knockout cells despite the reduction of H3K27Me3 by GSK126 (Supplementary Fig. 6h, i). Likewise, GSK126 did not increase the expression of the identified CARM1-regulated EZH2/BAF155

target genes in tumors formed by CARM1 knockout cells (Supplementary Fig. 6j). Together, these data support a model that EZH2 inhibition suppresses proliferation and promotes apoptosis in CARM1-expressing tumors by upregulating the EZH2/BAF155 target genes through regulating the antagonism between EZH2 (PRC2) and BAF155 (SWI/SNF) (Fig. 6g).

## Discussion

Our data demonstrate a dependence of CARM1-expressing cells on EZH2 activity, reflecting the silencing of EZH2 target tumor suppressor genes in a CARM1-dependent manner. CARM1 regulates the antagonism between EZH2 and BAF155 to drive the silencing of EZH2/BAF155 target tumor suppressor genes, which promotes apoptosis and inhibits proliferation. Specifically, CARM1-mediated methylation of BAF155 leads to the switch from BAF155 to EZH2 at the promoters of the EZH2/BAF155 target genes. Inhibition of EZH2 activity by clinically applicable small molecule restored the expression of the EZH2/BAF155 target genes. Thus, CARM1 regulates the antagonism between the

BAF155-containing SWI/SNF complex and the EZH2-containing PRC2 complex by methylating BAF155. In addition, CARM1-mediated methylation of BAF155 leads to the distribution of BAF155 to the BAF155Me target genes such as *TIMP3* in an EZH2-independent manner. This suggests that CARM1 functions to promote the expression of BAF155Me target genes while silencing EZH2/BAF155 target genes through EZH2-mediated H3K27Me3. However, CARM1 does not affect EZH2 expression levels (Fig. 1e, f). Instead, CARM1 promotes EZH2's distribution to the EZH2/BAF155 target genes (Figs. 4, 5). The observed selectivity against CARM1 by EZH2 inhibition is not due to variation in cell proliferation (e.g., Figs. 1a, d, 2a). Interestingly, a recent study shows that SWI/SNF opposes PRC2 through its rapid and ATP-dependent eviction[32]. Thus, it is possible that methylation of BAF155 by CARM1 suppresses the eviction of EZH2-containing PRC2 complex by BAF155-containing SWI/SNF complex. Together, these findings support that the observed selectivity against CARM1 by EZH2 inhibition is due to CARM1's role in regulating EZH2's association with the EZH2/BAF155 target genes without directly affecting EZH2 expression.

CARM1 plays a context-dependent role in cancer. Whereas the prevailing data support an overall oncogenic role of CARM1 in cancers, emerging evidence indicates that CARM1 may also positively regulate the activity of tumor suppressors[33] and promote the expression of tumor suppressor genes such as *TIMP3* through BAF155Me[6]. Future studies using genetically engineered mouse models will be informative in determining the role of CARM1 in different context. Regardless, directly targeting CARM1 may have unintended tumor-promoting effects. In addition, CARM1 is specifically required for postnatal survival[7]. Together, these caveats suggest that directly targeting CARM1 may not be a valid therapeutic strategy. In contrast, our data clearly demonstrate that EZH2 inhibition can suppress the growth of CARM1-expressing tumors and improves survival of tumor bearing mice. Thus, targeting EZH2 activity may be advantageous compared to inhibition of CARM1 activity.

Analysis of HGSOC patients from TCGA revealed that *CARM1* is amplified in ~10% and overexpressed in an additional ~10% of spontaneous HGSOC[19]. In comparison, somatic *BRCA1/2* mutations occur in ~3–4% of these cases for each gene that are among the most commonly mutated genes in HGSOC[13]. Interestingly, *CARM1* amplification does not typically occur in HGSOC with mutations in *BRCA1/2*. Thus, there is an even greater need for developing therapeutic approaches that correlate with CARM1 status. This is because platinum-based chemotherapy, the current standard of care, and emerging treatment with PARP inhibitors are typically more effective in patients with *BRCA1/2* inactivation[34]. In addition, there is no evidence that subunits of the SWI/SNF complex are mutated in HGSOC[19]. Likewise, there is no report to suggest that the cell lines we used in the current study carry mutations in the SWI/SNF subunits. Thus, CARM1 overexpression and/or amplification may serve as a predictive marker for further development of EZH2 inhibitor as a potential therapy in ovarian cancer.

In summary, our studies demonstrate that targeting EZH2 methyltransferase activity through the use of EZH2 inhibitors in CARM1-expressing cells represents a novel therapeutic strategy. Notably, EZH2 inhibitors such as GSK126 are well tolerated with limited toxicity in clinical trials for hematopoietic malignancies[17]. Thus, our studies provide scientific rationale for potential translation of these findings by repurposing the clinically applicable EZH2 inhibitors for CARM1-expressing EOCs. Given that CARM1 overexpression is frequently observed in many different cancer types[2–4], our findings may have far-reaching implications for improving therapy for an array of cancer types.

## Methods

**Cell lines and culture conditions**. The protocol for using primary cultures of human high-grade serous ovarian tumor cells was approved by The Wistar Institute/Christiana Care Health System Institutional Review Board. Human EOC cell lines were obtained from ATCC or Japanese Collection of Research Bioresources and were re-authenticated by The Wistar Institute's Genomics Facility at the end of experiments within last 3 months using short tandem repeat profiling using AmpFLSTR Identifiler PCR Amplification kit (Life Technologies) and cultured as previously described[15]. Human fallopian tube epithelial cells were obtained from Dr. Ron Drapkin. HOSE cells were in MCDB-105:199 (1:1) media with 15% FBS, 10 ng/ml EGF, 5 mg/ml Insulin, 100 mg/ml Bovine Pituitary Exract, and 500 ng/ml Hydrocortisone[15]. Mycoplasma testing was performed by LookOut Mycoplasma PCR detection (Sigma).

**Reagents and antibodies**. Small molecules used in the epigenetic screen were obtained from Structural Genomics Consortium or The Wistar Institute Molecular Screening Facility. GSK126 was obtained from Active Biochem or Xcess Biosciences. Antibodies were obtained from: mouse anti-CARM1 (Cell Signaling, Cat. No: 12495, 1:1000 for immunoblotting), goat anti-BAF155 (Santa Cruz, Cat. No: SC9746, 1:1000 for immunoblotting), rabbit anti-methylated R1064 BAF155 (Millipore, Cat. No: ABE1339, 1:1000 for immunoblotting), rabbit anti-EZH2 (Cell Signaling, Cat. No: 5246, 1:1000 for immunoblotting), rabbit anti-cleaved PARP p85 (Promega, Cat. No: G7341, 1:1000 for immunoblotting), mouse anti-Ki67 (Cell Signaling, Cat. No: 9449, 1:500 for IHC), rabbit anti-cleaved caspase 3 (Cell Signaling, Cat. No: 9661, 1:1000 for immunoblotting and 1:50 for IHC), rabbit anti-H3K27Me3 (Cell Signaling, Cat. No: 9733, 1:1000 for immunoblotting and 1:100 for IHC), mouse anti-β-actin (Sigma, Cat. No: A1978, 1:20,000 for immunoblotting), rabbit anti-RNA pol II (Santa Cruz, Cat. No: sc-899). Growth factor reduced basement membrane matrix (Matrigel) was obtained from Corning. Unprocessed scans of blots are available in the Supplementary Fig. 7.

**CRISPR-mediated CARM1 knockout**. pLentiCRISPR-CARM1 was constructed by inserting the CARM1 guide RNA (gRNA; 5′-AGCACGGAAAATCTACGC GG-3′)[35]. pLentiCRISPR v2 (Addgene) was digested and dephosphorylated with *BsmBI* restriction enzyme (Fermentas) for 30 min at 37 °C. The digested plasmid was run on a 1% agarose gel, cut out, and purified using the Wizard SV Gel and PCR Clean Up kit (Promega). The oligonucleotides were phosphorylated using T4 PNK (M0201S) with T4 Ligation Buffer (New England Biolabs, Inc.). Samples were annealed at a thermocycler at 37 °C for 30 min and then at 95 °C for 5 min and then were ramped down to 25 °C at 5 °C/min. Annealed oligonucleotides were diluted 1:200 in RNase/DNase-free water. Ligation of the annealed oligonucleotide and digested pLentiCRISPR v2 plasmid was performed using Quick Ligase (New England Biolabs, Inc.).

**Lentivirus and retrovirus infection**. Retrovirus production and transduction were performed using Phoenix cells (a gift of Dr. Gary Nolan, Stanford University)[23,36]. Lentivirus was packaged using the Virapower Kit from Invitrogen according to the manufacturer's instructions as described previously[15,37]. The following shRNAs obtained from the Molecular Screening Facility at The Wistar Institute were used: pLKO.1-shCARM1 (TRCN0000059090 and TRCN0000059090), pLKO.1-shEZH2 (TRCN0000040073), and pLKO.1-shBAF155 (TRCN00001353636). Cells infected with viruses encoding the puromycin resistance gene were selected in 1 μg/ml puromycin.

For ectopic CARM1 expression, lentiviral EX-Y3476-Lv105 encoding human CARM1 expression construct was obtained from Genecopoeia and lentivirus production was performed following the manufacturer's instruction using Lenti-Pac expression packaging kit (Genecopoeia).

**Reverse-transcriptase quantitative PCR**. RNA was isolated by RNeasy Mini Kit (Qiagen). mRNA relative expression for *DAB2*, *DLC1*, and *PMAIP* was determined using SYBR green 1-step iScript (Bio-Rad) with a Life Technologies QuantStudio 3. The primers were: 5′-TTCATTGCCCGTGATGTGACA-3′ (*DAB2* forward) and 5′-CCTGTTGCCCGGTTTTTATGG-3′ (*DAB2* reverse); 5′-AACCCAA-GACTACGGCTATTCA-3′ (*DLC1* forward) and 5′-CATAAAGCTGTGCA-TACTGGGG-3′ (*DLC1* reverse); 5′-ACCAAGCCGGATTTGCGATT-3′ (*NOXA* forward) and 5′-ACTTGCACTTGTTCCTCGTGG-3′ (*NOXA* reverse); and 5′-CATGTGCAGTACATCCATACGG-3′ (*TIMP3* forward) and 5′-CATCATA-GACGCGACCTGTCA-3′ (*TIMP3* reverse).

**Annexin V assay**. Phosphatidylserine externalization was detected using an Annexin V staining kit (Millipore) following the manufacturer's instructions. Annexin V-positive cells were detected using the Guava System and analyzed with the Guava Nexin Software Module (Millipore).

**3D Matrigel assays**. Matrigel was coated on the bottom of 8-well chamber slides and cells were plated on the Matrigel (4000 cells/well) in a 3% Matrigel/Media mixture. Media, Matrigel, and treatment (drug/vehicle) were replenished every

fourth day. On day 12, five bright-field images were captured from each well/treatment. Acini diameter was measured from images with ImageJ software (NIH).

**Colony formation assay.** 500–5000 cells were plated into a 24-well tissue culture plate and treated with the indicated compounds. Medium was changed every 3 days with appropriate drug doses for 12 days or until control wells became confluent. Colonies were washed twice with PBS and fixed with 10% methanol and 10% acetic acid in distilled water. Fixed colonies were stained with 0.005% crystal violet. Integrated density was measured using NIH ImageJ software.

**Intrabursal orthotopic xenograft models in vivo.** The protocols were approved by the Institutional Animal Care and Use Committee (IACUC). For in vivo experiments, the sample size of five mice per group was determined based on the data shown from in vitro experiments. Intrabursal orthotopic xenograft was performed[23,38]. $1 \times 10^6$ A1847 parental or A1847 CARM1 knockout cells were unilaterally injected into the ovarian bursa of 6–8-week-old female immunocompromised NSG mice ($n = 5$ per group). One week after injection the mice were randomized into two groups and treated with vehicle control (20% captisol) or GSK126 (50 mg/kg daily) for 3 weeks. At end of the experiments, tumors were surgically dissected and tumor burden was calculated based on tumor weight. For survival experiments, after stopping the treatment, the mice were followed for mortality or when the tumor burden reached 10% of body weight as determined by The Wistar Institute IACUC guideline.

**Subcutaneous xenograft and PDX models in vivo.** The protocols were approved by the IACUC. $5 \times 10^6$ control A1847 or A1847 CARM1 knockout cells were unilaterally injected subcutaneously into 6–8-week-old female immunocompromised NSG mice ($n = 5$ per group). One week after injection the mice were randomized and treated with vehicle control (20% captisol) or GSK126 (50 mg/kg daily). Tumor size was measured every 3 days for 3 weeks. Tumor size measurement was performed blindly but not randomly. At end of the experiments, tumors were surgically dissected and tumor burden was calculated based on tumor weight.

Similarly, $5 \times 10^6$ of control OVCAR3 or OVCAR3 CARM1-overexpressing cells were unilaterally injected subcutaneously into 6–8-week-old female immunocompromised NSG mice ($n = 5$ per group). One week after injection the mice were randomized and treated with vehicle control (20% captisol) or GSK126 (50 mg/kg daily). Tumor size was measured every 3 days for 4 weeks.

PDXs were generated using viably frozen stocks (in 10% DMSO) of second passage PDXs of HGSOC specimens established in ref. [39]. Two passage 3 HGSOC PDXs were re-established in NSG mice: TB776 and TB315. Specifically, TB776 or TB315 were unilaterally engrafted subcutaneously into 6–8-week-old female immunocompromised NSG mice ($n = 5$ per group). Mice were randomized and treated with vehicle control (20% captisol) or GSK126 (50 mg/kg daily) starting the day of the injection. Tumor size was measured every 3 days for 7 weeks.

**Epigenetic targeting small molecule set screen.** A1847 parental and CARM1 knockout cells were plated in 24-well plates and treated with 23 epigenetic compounds. Cell medium was changed every 3 days with appropriate drug doses for 14 days or until control wells became confluent. Colonies were washed twice with PBS and fixed with 10% methanol and 10% acetic acid in distilled water. Fixed colonies were stained with 0.005% crystal violet. Integrated density was measured using NIH ImageJ software as a surrogate for cell growth.

**RNA sequencing (RNA-Seq) and chromatin immunoprecipitation followed by sequencing (ChIP-seq).** RNA was extracted with Trizol (Invitrogen) and subsequently cleaned and DNase treated using RNeasy columns (Qiagen). DNase-treated RNA was subjected to library preparation. Libraries for RNA-Seq were prepared with ScriptSeq complete Gold kit (Epicenter) and subjected to a 75 bp paired-end sequencing run on NextSeq 500, using Illumina's NextSeq 500 high output sequencing kit following the manufacturer's instructions.

For ChIP-seq, cells were cross-linked with 1% formaldehyde for 10 min, followed by quenching with 125 mM glycine for 5 min. Fixed cells were resuspended in cell lysis buffer (10 mM Tris-HCl, pH 7.5, 10 mM NaCl, 0.5% NP-40) and incubated on ice for 10 min. The lysates were washed with MNase digestion buffer (20 mM Tris-HCl, pH 7.5, 15 mM NaCl, 60 mM KCl, 1 mM CaCl$_2$) once and incubated for 20 min at 37 °C in the presence of 1000 Gel units of MNase (NEB, M0247S) in 250 μL reaction volume. After adding the same volume of sonication buffer (100 mM Tris-HCl, pH 8.1, 20 mM EDTA, 200 mM NaCl, 2% Triton X-100, 0.2% sodium deoxycholate), the lysates were sonicated for 5 min (30 sec-on/30 sec-off) in a Diagenode bioruptor and centrifuged at 15,000 rpm for 10 min. The cleared supernatant equivalent to $2–4 \times 10^6$ cells was incubated with 2.5 μg of anti-EZH2 antibody (Cell Signaling, Cat. No. 5246) or 2 μg anti-H3K27Me3 antibody (Cell Signaling, Cat. No. 9733) on a rocker overnight. Bound chromatin was eluted and reverse cross-linked at 65 °C overnight. For next-generation sequencing, ChIP-seq libraries were prepared from 10 ng of ChIP and input DNA with the Ovation Ultralow DR Multiplex system (NuGEN). The ChIP-seq libraries were sequenced in a 51 bp paired-end run using the Illumina HiSeq 2000.

**Chromatin immunoprecipitation.** The following antibodies were used to perform ChIP: anti-H3K27Me3 (Cell Signaling, Cat. No: 9733), anti-BAF155 (Santa Cruz, Cat. No: sc-9746), anti-RNA polymerase II (Santa Cruz, Cat. No: sc-899), or anti-EZH2 (Cell Signaling, Cat. No: 5246). An isotype-matched IgG was used as a negative control[40,41]. ChIP DNA was analyzed by quantitative PCR against the promoter or a non-peak negative control region (2 Kb upstream of transcription starting site) of the indicated genes using the following primers: *DAB2* peak Forward: 5′- GTGTTCGGGGAGAAGTCAAA-3′, *DAB2* peak Reverse: 5′- ACGGAT CTGTGAAACGAAGC-3′, *DAB2* non-peak Forward: 5′-CGGGTTCACGC-CATTCT-3′ and *DAB2* non-peak Reverse: 5′-CACAGTGAAACCCTGTCTCTAC-3′; *DLC1* peak Forward: 5′-AAAATTTCCAAGCGCCACTA-3′, *DLC1* peak Reverse: 5′- ACACCGCCTTCTACCTTCCT-3′, *DLC1* non-peak Forward: 5′-ACTCTGTCTTCGAGGAGGAAATA-3′ and *DLC1* non-peak Reverse: 5′-ATCAGTGCCTAGGAGGAGTTAG-3′; *NOXA* peak Forward: 5′-TATTGTGG-GAGGTGGGGATA-3′, *NOXA* peak Reverse: 5′- GGCCTGAAAACTTAC-GATGG-3′, *DLC1* non-peak Forward: 5′- GCATTTCAGGGTGCGTATTTG-3′ and *DLC1* non-peak Reverse: 5′- AAACCACTCCAGGCTCATTT-3′; PCR primers for the *TIMP3* promoter are: *TIMP3* Forward: 5′-ACTCCCCTACGCAAG-GATTC-3′ and *TIMP3* Reverse: 5′-CGTGTGAAGGCAGTTTGGTT-3′.

**Bioinformatics and statistical analysis.** RNA-Seq data was aligned using bow-tie2[42] against hg19 version of the human genome and RSEM v1.2.12 software[43] was used to estimate raw read counts and RPKM using Ensemble gtf tracks. EdgeR[44] was used to estimate significance of differential expression between KO and parental samples. Overall gene expression changes were considered significant if passed FDR < 5%, Fold > 3 thresholds. ChIP-seq data was aligned using bowtie[45] against hg19 version of the human genome and HOMER[46] was used to call significant peaks in parental vs. CARM1 knockout comparison using style histone option and peaks that passed FDR < 1% threshold were called significantly decreased in knockout cells. Genes that had significantly decreased in knockout EZH2 and H3K27Me3 peak were considered and overlapped with genes significantly upregulated in knockout cells. Significance of overlap was tested using hypergeometric test using 57,736 Ensemble genes as a population size. Gene set enrichment analysis of gene sets was done using QIAGEN's Ingenuity® Pathway Analysis software (IPA®, QIAGEN Redwood City, www.qiagen.com/ingenuity) using "Diseases & Functions" and "Upstream Analysis" options. Functions with at least 10 member genes that passed $P < 10^{-5}$ threshold and upstream regulators (transcription factors only) that passed $P < 10^{-10}$ and had a significantly predicted activation state (|Z| > 2) were considered. TCGA Agilent microarray expression data for 528 ovarian cancer (OV) samples with copy number variation calls was downloaded. Expression of all genes was tested for negative association with CARM1 expression using Spearman and Pearson correlation and results overlapped with CARM1 parental/KO data using Entrez gene ID. EZH2 and CARM1 expression data were tested for differences between samples with amplified CARM1 vs. non-amplified CARM1 using two sample Student's $t$-test. Statistical analyses were performed using GraphPad Prism 5 (GraphPad) for Mac OS. Quantitative data are expressed as mean ± SEM unless otherwise stated. Spearman's or Pearson's test was used to measure statistical correlation. For all statistical analyses, the level of significance was set at 0.05.

**Data availability.** The RNA-Seq and ChIP-seq data was submitted to the Gene Expression Omnibus (GEO) database and can be accessed using accession number: GSE95645. All other data supporting the findings of this study are available upon request.

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

## Acknowledgements

We thank Mr. Xiang Hua for technical assistance. Dr. Wei Xu from University of Wisconsin for the plasmid encoding BAF155 K1064R mutant. This work was supported by US National Institutes of Health/National Cancer Institute grants (R01CA160331, R01CA163377, and R01CA202919 to R.Z., K99CA194318 to B.G.B.), US Department of Defense (OC140632P1 and OC150446 to R.Z.), and an Ovarian Cancer Research Fund Alliance program project (to R.Z.). J.H.L., T.O., and the Mayo Clinic Center for Individualized Medicine Epigenomics Development Laboratory were supported by P01DK68055, R01DK58185, and the Mayo Clinic Center for Individualized Medicine. Support of Core Facilities was provided by Cancer Center Support Grant (CCSG) CA010815 to The Wistar Institute.

## Author contributions

S.K. designed experiments. S.K., H.Z., S.W., Y.Y., B.G.B., and J.-H.L. conducted experiments. S.K., H.Z., Y.Y., B.G.B., J.-H.L., A.V.K., K.S.G., H.Y., and T.O. analyzed data. R.D. contributed critical experimental materials. J.R.C.-G. and D.W.S. contributed to experimental design. R.Z. conceived the study and supervised experiments. S.K., T.O., and R.Z. wrote the manuscript.

## Additional information

**Competing interests:** The authors declare no competing financial interests.

