## [Peer Review File · Nature Communications]

Reviewers' comments:

Reviewer #1 (Remarks to the Author):

The manuscript submitted by Karakashev et al. described the effects of CARM1's on ovarian cancer cells. The study concept is considered new and the translational potential is appreciated. The results from this study expand our understanding of how to harness epigenetic or epigenomic alterations for targeted therapy in the future. There are a couple of comments that the authors can proceed to improve the overall quality of this study.

1. According to the authors' hypothesis, knockout of CARM1 should affect H3K27Me3 or EZH2 either at expression level or chromatin re-distribution, subsequently affecting its downstream targets. However, this was not evident in the data provided (Fig. 1 and Fig. 4b).
2. Although the study design was solid and the results appeared to support the conclusion made by the authors, very few somatic mutations or gene rearrangement have been reported for CARM1 so far, therefore it is difficult to justify this gene as a canonical driver oncogene. In other words, its role in tumorigenesis is unclear and may require sophisticated genetics approaches to explore. These may include a combination of knock-in approaches in human cell lines and genetically engineered murine models. The study interpretation largely depended on pharmacological inhibitor(s) which may not be adequate because of the potential off-target nature of a CARM1 inhibitor and its associated high toxicity.
3. The ChIP-seq data shown in Figure 4 are of relatively poor quality with high noise background, especially the signals related to H3K27me3. The authors should show p-value or read count information to demonstrate significance of the peaks.
4. The Gene Expression Omnibus (GEO) data provided, GSE95645, is not currently available for the reviewers to evaluate the quality of the data and validity of the bioinformatics analysis. The ChIP-seq data as presented in Fig. 4b missed some important controls (e.g. EZH2 ChIP-seq in KO cells, H3K27me ChIP-seq in KO cells).
5. The reported amplicon on chr19p is not novel as several resident genes have been reported as the driver genes within this amplicon.
6. According to the authors' hypothesis, knockout of CARM1 should affect H3K27Me3 or EZH2 either at expression level or chromatin re-distribution, subsequently affecting its downstream targets. However, this was not evident in the data provided (Fig. 1 and Fig. 4b).
7. It would be more informative to perform CARM1 ChIP-seq and overlay the CARM target list with the CARM1-transcriptionally regulated gene list derived from RNAseq.
8. Regarding specificity of the pharmacological inhibitor, GSK126, and the associated phenotypes:

In Figure 2b and Supplemental Figure 6d. GSK126 affects H3K27Me3 levels regardless of CARM1 expression, however, in cytotoxicity study or in target gene expression study (Figure 2a and Supplemental Figure 6d, respectively), GSK126 inhibitor targets CARM-high cells but not CARM-low cells. There is a disconnection between biochemical property and the observed phenotypes.
9. The authors may report some interesting link between a gene located on chr19p and EZH2. However, to claim whether CARM1 is an oncogene or not requires much more research effort with a better designed genetic knock-in approach. This needs to be shown, or at least discussed.
10. Fig. 1C: Data from all 23 individual epigenetic inhibitors should be presented by a table.

11. Supplementary Fig. 2a&2b: Statistical evaluation is needed to assess whether there is a difference between CRAM1 high & low cells.

12. To strengthen the notion promoted by the authors on the utility of GSK126 treatment based on CRAM1 expression levels, the authors should perform a study on an extensive panel of primary cell cultures. Based on the data provided, only two primary cultures of HGSOV have been performed (Supplementary Fig. 2e-g). To further improve clinical implication of the current study, patient-derived xenograft (PDX) cancer model is recommended to be employed.

Reviewer #2 (Remarks to the Author):

In the manuscript entitled "CARM1-expressing ovarian cancer depends on the histone methyltransferase EZH2 activity" Karakashev et al., propose that CARM1 expression predicts the susceptibility of ovarian cancer cells to EZH2 inhibitors. Mechanistically, they provide data supporting a model where Baf155 recruitment to shared target genes occludes EZH2 recruitment. CARM1 ameliorates this occlusion by methylating Baf155 which, in turn, prohibits Baf155 recruitment to chromatin, thereby permitting the recruitment of EZH2 and subsequent gene repression. Overall, the data is interesting, especially the idea of antagonism between EZH2 and Baf155. This concept is quite novel, and provides insight regarding the recruitment of EZH2 to target genes. How EZH2 recruitment to chromatin is facilitated or occluded has been a topic of much debate and the mechanism remains unclear. The Baf155 knockdown and add-back experiments with Wt and mutant proteins are well executed and look convincing. However, many key experiments are missing and much of the paper is dependent on data from a single cell line, A1847, which is often not the ideal model for the experiments being described. Much of the data needs to be extended beyond A1847 using both CARM1 expressing and non-expressing cells as a comparison. Thus, the conclusions of the paper, while of interest, are not strongly supported.

Major points for revision:

1. For the authors to support their claim that CARM1 expression predicts sensitivity to EZH2 inhibitors, several key experiments are essential.

A.) The data in figure 1d needs to be complemented with a set of experiments where CARM1 is ectopically expressed in non-expressing cells (OVCAR3, OVCAR5 or PEO4) to test for increased sensitivity to EZH inhibitors.

B.) Figure 2A is interesting, but preliminary. This shows a single high dose (10 micromolar, at a single end point). Dose responses are required here. Perhaps this single time point, at this particular dose is the only approach where sensitivity to EZH2 inhibitors in CARM1 expressing cells can be observed.

C.) There should be a dose curve carried out for figure 1D.

2. How sensitive are several CARM1 expressing cell lines, beyond A1847 to EZH2 knockdown (OVCAR10, PEO4)? How does this compare to CARM1-negative cells (OVCAR3, OVCAR5 or PEO4).

3. Throughout the paper, it should be shown within the figure which cell line is being utilized.

4. Related to figure 3, the author states, "Since CARM1 may promote EZH2-dependent gene silencing, we focused on the genomic loci that showed a decrease in association with EZH2/H3K27Me3 in CARM1 KO cells compared with controls".

- If CARM1 knockdown reduces EZH2 association with chromatin by allowing Baf155 to be recruited, then knocking down Baf155 in OVCAR3, OVCAR5 or PEO4 cells should lead to increased EZH2 association with target gene promoters. This should be examined. A single cell line is not sufficient to prove a model.

5. Some corrections need to be made to figure 3.
- A) For figure 3b., the scale (eg. # of reads), needs to be clearly labels on the Y axis of the ChIP-seq tracks.
 - B) For the DCL1 ChIP-seq track, why does there appear to be CARM1 knockout reads (pink dots) at the top of the EZH2 peak?
 - C) Is the scale for CARM1 KO and control lanes at the same scale?
6. A western for Dimethy-BAF155 should be carried out for all the cell lines shown in Fig.1A. Is there a clear correlation between Baf155 methylation and CARM1 expression?
7. A) The ChIP for BAF155 and EZH2 in figure 4 shows a weak pulldown efficiency. Multiple sets of primers surrounding this binding site should be used to show the specificity at the enrichment at the proximal promoter. These additional primers would serve as necessary negative controls.
- B) Is there other literature supporting the claim that Baf155 antagonizes EZH2 recruitment to target genes? By visualizing at HeLa ChIP-seq tracks (UCSC Genome Browser), peaks for Baf155 and EZH2 at Noxa and perhaps DAB2 can be observed. It also appears that the EZH2 antibody has a very low signal-to-noise ratio highlighting the need for further amplicons in the ChIP assays for EZH2.
8. The data shown in figure 5 are missing critical experiments necessary to prove the hypothesis presented by the authors. In A1847 cells, CARM1 allows EZH2-driven repression of target genes (DAB2, DLC1 and NOXA). The authors take the approach of knocking down Baf155 in these cells and adding-back WT of mutants to look at gene expression. However, there problems are associated with this figure.
- A) This is really not the ideal system for proving the proposed mechanism. Using OVCAR3, OVCAR5 or PEO4 cells, does knocking down BAF155 repress DAB2, DLC1 or NOXA?
 - B) Does knocking down BAF155 in OVCAR3, OVCAR5 or PEO4 cells displace EZH2 by ChIP?
 - C) Does ectopic expression of CARM1 displace BAF155 by ChIP, in OVCAR3, OVCAR5 or PEO4 cells?
9. The authors claim that BAF155 is responsible for blocking the transcriptional repression imposed by EZH2. If this is the case, does BAF155 knockdown ameliorate the increase in DCL2, NOXA and DAB2 transcription (mRNA levels) seen in response to EZH2 inhibitors as shown in Figure 4?
10. EZH2 inhibitors reduce tumor growth in many xenograft models. The data shown here are not really meaningful unless a comparison can be made between the efficacy of GSK126 against A1847 versus OVCAR3, OVCAR5 or PEO4. The ideal model would be OVCAR3, OVCAR5 or PEO4 vs one of these cell lines ectopically expressing CARM1, which should sensitize the cells. For the xenograft experiment, why wasn't quantification of tumor volume included?
11. For ChIP and in fact all experiments, do the error bars represent SEM or STD? Also, how were P values calculated?

Minor points for revision:

1. In the introduction, the authors state "Here we show that EZH2 inhibition is selective against CARM1 expression in epithelial ovarian cancer."
- This is an ambiguous, grammatically awkward sentence and should be clarified. What does ".. is selective against CARM1 expression ..." mean?
2. In the abstract the authors state "Mechanistically, CARM1 promotes EZH2-mediated silencing of EZH2/BAF155 target tumor suppressor genes by methylating BAF155 to alter the antagonism between EZH2 and BAF155."

-Again, this sentence is ambiguous. What precisely does “..alter the antagonism..” mean? Does it increase or decrease the antagonism?

3. For Supplemental figure 1. How is “High CARM1” defined for the Kaplan Meier plots? This data does not seem to be supported by other databases such as “KM Plotter”.

Reviewer #3 (Remarks to the Author):

In the manuscript by Karakashev et al, the authors investigate the epigenetic vulnerabilities in CARM1-expressing ovarian cancer. First, the authors show that CARM1-high ovarian cancer cells are hypersensitive to chemical inhibitors of EZH2. This effect was found to correlate with apoptosis induction. On a mechanistic level, the authors find that CARM1 promotes the silencing of EZH2 target genes via its effect on methylation of BAF155. Evidence is provided that methylation of BAF155 at R1064 leads to displacement of BAF155 from chromatin, thus providing a logical mechanism for the EZH2 addiction in CARM1-high cancers, since it has been well established that the BAF complex antagonizes the function of EZH2 on chromatin. Finally, the authors perform therapeutic trials in xenograft model to show that EZH2 inhibitors can extend the survival of CARM1-high tumor cells.

Collectively, I found this to be an exciting and rigorous study that leads to a clear, solid conclusion of high biomedical interest. This study features a wide range of experimental assays, including in vivo tumor models, small molecules, CRISPR-based knockouts, epigenomics, and point mutations of EZH2/BAF155. This is a high-quality study that leads to an important discovery for the field of cancer epigenetics. I am in support of publishing this study in Nature Communications without further delay.

Point-by-Point Response to The Reviewers Comments

We sincerely thank both the Reviewers and the Editors for the constructive and thoughtful review provided for our manuscript. We are grateful for their shared appreciation of our manuscript as “*The study concept is considered new and the translational potential is appreciated (Reviewer 1)*”, “*Overall, the data is interesting, especially the idea of antagonism between EZH2 and Baf155 (Reviewer 2)*”, and “*This is a high-quality study that leads to an important discovery for the field of cancer epigenetics (Reviewer 3)*”. All the comments raised are truly valuable to improve the manuscript. Correspondingly, we have strived to provide new experimental answers to their comments. I hope that there is no doubt that we have taken the Reviewers’ and Editors’ comments very seriously. We believe that by addressing the reviewers’ concerns we have produced a more solid and cohesive manuscript. A point-by-point response to the reviewers’ comments is detailed below with original comments italicized. Changes that directly address the reviewers’ concerns were denoted with vertical lines in the right margin in the revised manuscript. We hope the Reviewers and the Editors will find this manuscript to be much improved and suitable for publication.

Reviewers' comments:

Reviewer #1 (Remarks to the Author):

The manuscript submitted by Karakashev et al. described the effects of CARM1’s on ovarian cancer cells. The study concept is considered new and the translational potential is appreciated. The results from this study expand our understanding of how to harness epigenetic or epigenomic alterations for targeted therapy in the future. There are a couple of comments that the authors can proceed to improve the overall quality of this study.

1. *According to the authors’ hypothesis, knockout of CRAM1 should affect H3K27Me3 or EZH2*

either at expression level or chromatin re-distribution, subsequently affecting its downstream targets. However, this was not evident in the data provided (Fig. 1 and Fig. 4b).

Response: We thank the reviewer for the comments and apologize for the confusion due to the overlay presentation of the ChIP-seq tracks in control parental vs. CARM1 knockout cells in Figure 3b in the initial submission. As reviewer rightly pointed out, our data supports a model whereby CARM1 affects the chromatin re-distribution of EZH2. Indeed, as clearly demonstrated in **revised Figure 3b** with separate ChIP-seq tracks, CARM1 knockout significantly affects the EZH2 chromatin redistribution. This was validated in Figure 4b that showed that CARM1 knockout led to loss of EZH2 from the promoters of the pro-apoptotic BAF155/EZH2 target genes.

2. Although the study design was solid and the results appeared to support the conclusion made by the authors, very few somatic mutations or gene rearrangement have been reported for CARM1 so far, therefore it is difficult to justify this gene as a canonical driver oncogene. In other words, its role in tumorigenesis is unclear and may require sophisticated genetics approaches to explore. These may include a combination of knock-in approaches in human cell lines and genetically engineered murine models. The study interpretation largely depended on pharmacological inhibitor(s) which may not be adequate because of the potential off-target nature of a CARM1 inhibitor and its associated high toxicity.

Response: We thank that reviewer for the comments. As the reviewer rightly pointed out, CARM1 is not mutated or genetic rearranged in human cancers. Instead, CARM1 is amplified in cancers such as breast cancer ¹ and, as reported for the first time in this manuscript, in ovarian cancer. We agree with the reviewer that genetically engineered murine model is beyond the scope of the current study. However, our new data showed that ectopic expression

of CARM1 in CARM1-low epithelial ovarian cancer cells promotes tumor growth *in vivo* in xenograft models (**New Data Supplementary Figure 6d**). We apologize for the confusion, while it has been demonstrated that CARM1 inhibition is associated with high toxicity, the current study focused on EZH2 inhibitor in CARM1-expressing cells. We did not use CARM1 inhibitors in our studies. Notably, EZH2 inhibitor is well tolerated in early phase clinical trials ², which highlighted the novelty and significance of the present study.

3. *The ChIP-seq data shown in Figure 4 are of relatively poor quality with high noise background, especially the signals related to H3K27me3. The authors should show p-value or read count information to demonstrate significance of the peaks.*

Response: We thank the reviewer for the comment and apologize for the confusion caused by the overlay ChIP-seq data presentation in the initial submission. As showed in the revised manuscript (Figure 3), the ChIP-seq data is of high quality and showed a high signal to noise ratio. We have now included the read count as suggested by the reviewer.

4. *The Gene Expression Omnibus (GEO) data provided, GSE95645, is not currently available for the reviewers to evaluate the quality of the data and validity of the bioinformatics analysis. The ChIP-seq data as presented in Fig. 4b missed some important controls (e.g. EZH2 ChIP-seq in KO cells, H3K27me ChIP-seq in KO cells).*

Response: Again, we apologize for the confusion caused by the overlay ChIP-seq data presentation for Figure 3b in the initial submission. We have now separated all the tracks with all the controls such as EZH2 ChIP-seq in KO cells and H3K27Me3 ChIP-seq in KO cells clearly indicated on separate tracks in **revised Figure 3b**. We apologize for the GEO data access and have now uploaded the dataset on our server. Direct download link for the data

https://genomics.wistar.upenn.edu/Data/Rugang/GeoSubmission_Priyankara_02282017.tar.gz

Additional sample and experiment information can be seen in GEO:

<https://www.ncbi.nlm.nih.gov/geo/query/acc.cgi?token=gpsbemsbrszlyp&acc=GSE95645>

5. *The reported amplicon on chr19p is not novel as several resident genes have been reported as the driver genes within this amplicon.*

Response: We thank the reviewer for the comments. While we agree that amplicon on chr19p has been reported, to the best of knowledge this is the first study that links *CARM1* amplification to ovarian cancer. Significantly, we discovered a novel *CARM1* regulated antagonism between BAF155 and EZH2, which underlies the observed sensitivity to EZH2 inhibitor in a *CARM1* status dependent manner.

6. *According to the authors' hypothesis, knockout of CRAM1 should affect H3K27Me3 or EZH2 either at expression level or chromatin re-distribution, subsequently affecting its downstream targets. However, this was not evident in the data provided (Fig. 1 and Fig. 4b).*

Response: We thank the reviewer for the comments and apologize for the confusion due to the overlay ChIP-seq data presentation in Figure 3b in the initial submission. As detailed in response to Point #1 above, our data supports a model whereby *CARM1* affects the chromatin re-distribution of EZH2. Indeed, as clearly demonstrated in revised Figure 3b with separate ChIP-seq tracks, *CARM1* knockout significantly affects the EZH2 chromatin redistribution, which was validated in the ChIP analysis in Figure 4b.

7. *It would be more informative to perform CARM1 ChIP-seq and overlay the CARM target list with the CARM1-transcriptionally regulated gene list derived from RNAseq.*

Response: We thank the reviewer for the comments. We did try CARM1 ChIP initially. Consistent with previous reports³, we were unable to ChIP CARM1. This is consistent with our data that supports a model whereby CARM1 regulates BAF155/EZH2 target gene expression through methylating BAF155. Thus, CARM1 may not directly bind to chromatin³, instead it regulates these gene expression by methylating BAF155 that affects EZH2 binding to chromatin.

8. *Regarding specificity of the pharmacological inhibitor, GSK126, and the associated phenotypes:*

In Figure 2b and Supplemental Figure 6d. GSK126 affects H3K27Me3 levels regardless of CARM1 expression, however, in cytotoxicity study or in target gene expression study (Figure 2a and Supplemental Figure 6d, respectively), GSK126 inhibitor targets CRAM1-high cells but not CARM1-low cells. There is a disconnection between biochemical property and the observed phenotypes.

Response: We thank the reviewer for the comment and again apologize for the confusion cause by the overlay presentation of ChIP-seq data in the initial submission. As the reviewer rightly pointed out, our data supports that CARM1 affects EZH2 chromatin redistribution. Only in CARM1-high cells, EZH2 is distributed to the EZH2/BAF155 target tumor suppressor genes. When EZH2 activity is inhibited, these cells undergo apoptosis. However, in CARM1-low cells, although EZH2 inhibitor can reduce H3K27Me3 equally efficient, these cells will not respond because EZH2 is not present in these pro-apoptotic genes. Thus, the observed biochemical property and the observed phenotypes are consistent with our model.

9. *The authors may report some interesting link between a gene located on chr19p and EZH2.*

However, to claim whether CRAM1 is an oncogene or not requires much more research effort with a better designed genetic knock-in approach. This needs to be shown, or at least discussed.

Response: We agree with the reviewer and please see response to Point #2 above. In addition, as requested, we have now added the relevant discussion on page 15, paragraph 2.

10. *Fig. 1C: Data from all 23 individual epigenetic inhibitors should be presented by a table.*

Response: We thank the reviewer for the comments. The data is included in supplementary Table 1.

11. *Supplementary Fig. 2a&2b: Statistical evaluation is needed to assess whether there is a difference between CARM1 high & low cells.*

Response: We thank the reviewer for the comments. As requested, we have now performed the statistical evaluation. The difference between CARM1 high vs. low tumor specimens in Supplemental Figure 2a is significant. While same trend was observed in Supplemental Figure 2b, the statistical analysis did not reach significance. This is likely due to the limited number of tumor cases with low CARM1 expression (n=3). Regardless, although we agree the reviewer that these results are informative, as the reviewer pointed out, our data showed that CARM1 does not affect EZH2 expression *per se*, but instead affects the chromatin redistribution of EZH2. Consistently, EZH2 expression is comparable in the TCGA ovarian cancer dataset with or without *CARM1* amplification (Supplementary Figure 1f).

12. *To strengthen the notion promoted by the authors on the utility of GSK126 treatment based*

on CARM1 expression levels, the authors should perform a study on an extensive panel of primary cell cultures. Based on the data provided, only two primary cultures of HGSOC have been performed (Supplementary Fig. 2e-g). To further improve clinical implication of the current study, patient-derived xenograft (PDX) cancer model is recommend to be employed.

Response: We agree with the reviewer for the comments. Within the limited time during revision, we were only able to obtain one additional primary ovarian tumor. However, we were unsuccessful in obtaining primary culture from this tumor (data not shown). Significantly, as requested, we have now performed the experiments using PDX models derived from tumors different from those used for primary cultures showed in the manuscript and showed that EZH2 inhibitor GSK126 significantly suppressed the growth of CARM1-high, but not CARM1-low, ovarian cancer PDXs (**New Data Supplementary Figure 6e-g**).

Reviewer #2 (Remarks to the Author):

In the manuscript entitled “CARM1-expressing ovarian cancer depends on the histone methyltransferase EZH2 activity” Karakashev et al., propose that CARM1 expression predicts the susceptibility of ovarian cancer cells to EZH2 inhibitors. Mechanistically, they provide data supporting a model where Baf155 recruitment to shared target genes occludes EZH2 recruitment. CARM1 ameliorates this occlusion by methylating Baf155 which, in turn, prohibits Baf155 recruitment to chromatin, thereby permitting the recruitment of EZH2 and subsequent gene repression. Overall, the data is interesting, especially the idea of antagonism between EZH2 and Baf155. This concept is quite novel, and provides insight regarding the recruitment of EZH2 to target genes. How EZH2 recruitment to chromatin is facilitated or occluded has been a topic of much debate and the mechanism remains unclear. The Baf155 knockdown and add-back experiments with Wt and mutant proteins are well executed and look convincing. However, many key experiments are missing and much of the paper is dependent on data from

a single cell line, A1847, which is often not the ideal model for the experiments being described. Much of the data needs to be extended beyond A1847 using both CARM1 expressing and non-expressing cells as a comparison. Thus, the conclusions of the paper, while of interest, are not strongly supported.

Major points for revision:

1. For the authors to support their claim that CARM1 expression predicts sensitivity to EZH2 inhibitors, several key experiments are essential.

A.) The data in figure 1d needs to be complimented with a set of experiments where CARM1 is ectopically expressed in non-expressing cells (OVCAR3, OVCAR5 or PEO4) to test for increased sensitivity to EZH2 inhibitors.

B.) Figure 2A is interesting, but preliminary. This shows a single high dose (10 micromolar, at a single end point). Dose responses are required here. Perhaps this single time point, at this particular dose is the only approach where sensitivity to EZH2 inhibitors in CARM1 expressing cells can be observed.

C.) There should be a dose curve carried out for figure 1D.

Response: We thank the reviewer for these important comments. We have now performed the requested experiments and our new results showed that CARM1 ectopic expression increases BAF155Me, which correlates with an increase in the sensitivity to the EZH2 inhibitor in CARM1-low OVCAR3 and CAOV3 cells (**New Data Figure 2c-d and Supplementary Fig. 2k-l**). As requested, we have now performed the dose responses. Our new results support the notion that compared with either normal fallopian tube/ovarian surface epithelial cells or CARM1 low-expressing cells, CARM1 high-expressing cells are more sensitive to the EZH2 inhibitor (**New**

Data Supplementary Fig. 2e). And we have now included the dose-curves for GSK126 in A1847 and CARM1 knockout cells (New Data Supplementary Fig. 1k)

Figure 1 for Reviewer. **a**, Expression of EZH2 and a loading control b-actin the indicated CARM1-high and CARM1-low cell lines. **b**, Same as a, but the cells were subjected to colony formation assay. * $P < 0.001$. Mean of three independent experiments with SEM.

negative cells (OVCAR3, OVCAR5 or PEO4).

2. How sensitive are several CARM1 expressing cell lines, beyond A1847 to EZH2 knockdown (OVCAR10, PEO4)? How does this compare to CARM1-

Response: We thank the reviewer for the comments. As we and others published, EZH2 knockdown suppresses the growth of EZH2-expressing cells^{4, 5, 6}. This occurs regardless of CARM1 status (**Figure 1 for reviewer**). This is likely due to PRC2-independent function of EZH2⁵. However, our current study established that CARM1-high cells are more sensitive to inhibition of EZH2 methyltransferase activity. As such, compared with EZH2 expression, the current study established a therapeutic opportunity for CARM1-high expressing cells by targeting EZH2's methyltransferase activity using clinical applicable EZH2 inhibitors.

3. Throughout the paper, it should be shown within the figure which cell line is being utilized.

Response: We thank the reviewer for the comments. This has now been done.

4. Related to figure 3, the author states, "Since CARM1 may promote EZH2-dependent gene silencing, we focused on the genomic loci that showed a decrease in association with

EZH2/H3K27Me3 in CARM1 KO cells compared with controls”.

- If CARM1 knockdown reduces EZH2 association with chromatin by allowing Baf155 to be recruited, then knocking down Baf155 in OVCAR3, OVCAR5 or PEO4 cells should lead to increased EZH2 association with target gene promoters. This should be examined. A single cell line is not sufficient to prove a model.

Response: We thank the reviewer for the comments. We have now performed the experiment. Our new data showed that BAF155 knockdown increased EZH2's association with target gene promoters in CARM-low OVCAR3 and CAOV3 cells (**New Data Figure 4f**).

5. *Some corrections need to be made to figure 3.*

A) For figure 3b., the scale (eg. # of reads), needs to be clearly labels on the Y axis of the ChIP-seq tracks.

Response: We thank the reviewer for the comments. We have now labeled the scales as suggested on revised Figure 3b.

B) For the DCL1 ChIP-seq track, why does there appear to be CARM1 knockout reads (pink dots) at the top of the EZH2 peak?

Response: We apologize for the confusion. The ChIP-seq tracks were overlays between control (yellow) and CARM knockout cells (purple). We have now separated the tracks for clarity in revised Figure 3b.

C) Is the scale for CARM1 KO and control lanes at the same scale?

Response: Again, we apologize for using the overlay in data presentation in the initial submission. All the scales for CARM KO and control tracks were the same. To avoid confusion, we have now separated all the overlay tracks.

6. *A western for Dimethy-BAF155 should be carried out for all the cell lines shown in Fig.1A. Is there a clear correlation between Baff155 methylation and CARM1 expression?*

Response: We agree with the reviewer and have now performed the suggested experiments. Overall, there is a correlation between BAF155 methylation and CARM1 expression (**New Data Supplementary Figure 4a**).

7. *A) The ChIP for BAF155 and EZH2 in figure 4 shows a weak pulldown efficiency. Multiple sets of primers surrounding this binding site should be used to show the specificity at the enrichment at the proximal promoter. These additional primers would serve as necessary negative controls.*

Response: We thank the reviewer for the comments. As suggested, we have now included a negative control in non-peak region and an additional peak promoter region to show that changes in BAF155 and EZH2 are peak specific (**New Data Supplementary Figure 4f-g**).

B) Is there other literature supporting the claim that Baf155 antagonizes EZH2 recruitment to target genes? By visualizing at HeLa ChIP-seq tracks (UCSC Genome Browser), peaks for Baf155 and EZH2 at Noxa and perhaps DAB2 can be observed. It also appears that the EZH2 antibody has a very low signal-to-noise ratio highlighting the need for further amplicons in the ChIP assays for EZH2.

Response: We thank the reviewer for the comments. To the best of our knowledge, the findings reported in the current manuscript for the first time established the antagonism between BAF155 and EZH2. We are very glad that mining of other independent databases supports our conclusion. Again, we apologize for the confusion with the overlay tracks in the initial submission which were mistaken for comparison. As showed in the revised Figure 3b, the EZH2 ChIP-seq showed a very high signal-to-noise ratio (e.g., CARM1 knockout leads to nearly complete loss of EZH2 binding).

8. *The data shown in figure 5 are missing critical experiments necessary to prove the hypothesis presented by the authors. In A1847 cells, CARM1 allows EZH2-driven repression of target genes (DAB2, DLC1 and NOXA). The authors take the approach of knocking down Baf155 in these cells and adding-back WT of mutants to look at gene expression. However, there problems are associated with this figure.*

A) This is really not the ideal system for proving the proposed mechanism. Using OVCAR3, OVCAR5 or PEO4 cells, does knocking down BAF155 repress DAB2, DLC1 or NOXA?

B) Does knocking down BAF155 in OVCAR3, OVCAR5 or PEO4 cells displace EZH2 by ChIP?

C) Does ectopic expression of CARM1 displace BAF155 by ChIP, in OVCAR3, OVCAR5 or PEO4 cells?

Response: We thank the reviewer for the suggestion. We have now performed the requested experiment. In CARM1-low OVCAR3 and CAOV3 cells, our new data showed that knocking down BAF155 repressed DAB2, DLC1 and NOXA transcription as determined by qRT-PCR (**New Data Supplementary Figure 4h**) and displaced EZH2 as determined by ChIP (**New Data Figure 4f**), and ectopic expression of CARM1 displaced BAF155 as determined by ChIP (**New Data Figure 4e**).

9. *The authors claim that BAF155 is responsible for blocking the transcriptional repression imposed by EZH2. If this is the case, does BAF155 knockdown ameliorate the increase in DCL2, NOXA and DAB2 transcription (mRNA levels) seen in response to EZH2 inhibitors as shown in Figure 4?*

Response: We would like to clarify that in CARM1-high cells, BAF155 is displaced by EZH2 to silence DCL1, NOXA and DAB2. Consequently, EZH2 inhibition increases the expression of these target genes. Thus, BAF155 knockdown will decrease the expression of these genes only in the CARM1-low cells. Consistently, our new results showed that BAF155 knockdown decreased the expression of DAB2, DLC1 and NOXA in CARM1-low OVCAR3 and CAVO3 cells (**New Data Supplementary Figure 4h**).

10. *EZH2 inhibitors reduce tumor growth in many xenograft models. The data shown here are not really meaningful unless a comparison can be made between the efficacy of GSK126 against A1847 versus OVCAR3, OVCAR5 or PEO4. The ideal model would be OVCAR3, OVCAR5 or PEO4 vs one of these cell lines ectopically expressing CARM1, which should sensitize the cells. For the xenograft experiment, why wasn't quantification of tumor volume included?*

Response: We thank the reviewer for the comments. We have now performed the suggested experiments. Our new data show that CARM1-low OVCAR3 failed to response to GSK126, while CARM1 ectopically expressing OVCAR3 cells responded to GSK126 (**New Data Supplementary Figure 6d**). For the xenograft experiment, tumor volume was qualified in supplementary Figure 6b.

11. *For CHIP and in fact all experiments, do the error bars represent SEM or STD? Also, how were P values calculated?*

Response: We thank the reviewer for the comments. Error bar represents SEM for all the experiments unless otherwise stated in the figure legends. Two-tailed paired *t*-test was used to calculate *P*-values. We have now included these information in the figure legends.

Minor points for revision:

1. *In the introduction, the authors state “Here we show that EZH2 inhibition is selective against CARM1 expression in epithelial ovarian cancer.”*

- This is an ambiguous, grammatically awkward sentence and should be clarified. What does “.. is selective against CARM1 expression ...” mean?

Response: We thank the reviewer for the comments. We have now changed the statement to “Here we show that EZH2 inhibition is effective in CARM1-expressing epithelial ovarian cancer.”

2. *In the abstract the authors state “Mechanistically, CARM1 promotes EZH2-mediated silencing of EZH2/BAF155 target tumor suppressor genes by methylating BAF155 to alter the antagonism between EZH2 and BAF155.”*

-Again, this sentence is ambiguous. What precisely does “..alter the antagonism..” mean? Does it increase or decrease the antagonism?

Response: We agree with the reviewer. We have now changed the sentence to “Mechanistically, CARM1 promotes EZH2-mediated silencing of EZH2/BAF155 target tumor suppressor gene by methylating BAF155, which leads to the displacement of BAF155 by EZH2.

3. *For Supplemental figure 1. How is “High CARM1” defined for the Kaplan Meier plots? This data does not seem to be supported by other databases such as “KM Plotter”.*

Response: We thank the reviewer for the comments. We used *CARM1* amplified vs. not amplified in the most well-annotated and carefully quality-controlled TCGA dataset for Kaplan Meier plots. For expression, we used a continuous expression model to statistically determine the cutoff expression levels for high vs. low expression. However, we did not check other databases. To avoid confusion in the literature, we have now removed the expression panel from supplemental Figure 1.

Reviewer #3 (Remarks to the Author):

In the manuscript by Karakashev et al, the authors investigate the epigenetic vulnerabilities in CARM1-expressing ovarian cancer. First, the authors show that CARM1-high ovarian cancer cells are hypersensitive to chemical inhibitors of EZH2. This effect was found to correlate with apoptosis induction. On a mechanistic level, the authors find that CARM1 promotes the silencing of EZH2 target genes via its effect on methylation of BAF155. Evidence is provided that methylation of BAF155 at R1064 leads to displacement of BAF155 from chromatin, thus providing a logical mechanism for the EZH2 addiction in CARM1-high cancers, since it has been well established that the BAF complex antagonizes the function of EZH2 on chromatin. Finally, the authors perform therapeutic trials in xenograft model to show that EZH2 inhibitors can extend the survival of CARM1-high tumor cells.

Collectively, I found this to be an exciting and rigorous study that leads to a clear, solid conclusion of high biomedical interest. This study features a wide range of experimental assays, including in vivo tumor models, small molecules, CRISPR-based knockouts, epigenomics, and point mutations of EZH2/BAF155. This is a high-quality study that leads to an important discovery for the field of cancer epigenetics. I am in support of publishing this study in Nature Communications without further delay.

Response: We are very appreciative for the positive comments by the reviewer for our efforts in addressing the initial comments raised at Nature Medicine before transferring to Nature Communications.

References:

1. Yang Y, Bedford MT. Protein arginine methyltransferases and cancer. *Nat Rev Cancer* **13**, 37-50 (2013).
2. Ribrag V, *et al.* Phase 1 first-in-human study of the enhancer of zeste-homolog 2 (EZH2) histone methyl transferase inhibitor E7438 as a single agent in patients with advanced solid tumors or B cell lymphoma. *Eur J Cancer* **50**, 197-197 (2014).
3. Wang L, *et al.* CARM1 methylates chromatin remodeling factor BAF155 to enhance tumor progression and metastasis. *Cancer Cell* **25**, 21-36 (2014).
4. Wang Y, *et al.* The histone methyltransferase EZH2 is a therapeutic target in small cell carcinoma of the ovary, hypercalcaemic type. *J Pathol* **242**, 371-383 (2017).

5. Kim KH, *et al.* SWI/SNF-mutant cancers depend on catalytic and non-catalytic activity of EZH2. *Nature medicine* **21**, 1491-1496 (2015).

6. Li H, Cai Q, Godwin AK, Zhang R. Enhancer of zeste homolog 2 promotes the proliferation and invasion of epithelial ovarian cancer cells. *Molecular cancer research : MCR* **8**, 1610-1618 (2010).

REVIEWERS' COMMENTS:

Reviewer #1 (Remarks to the Author):

The authors have adequately addressed the prior concerns. There are no more questions from my end.

Reviewer #2 (Remarks to the Author):

In their revised manuscript, the authors have now completed all of the requested experiments (quite a feat considering the short period of time required for revision). In my opinion, the work is now complete enough, and includes the proper controls, for publication.

A couple of minor textual revisions should be addressed before publication.

#1. In their haste to prepare figures, it seems the Control and knockdown were switched in Supplemental figure 4h. As it stands, it appears Baf155 knockdown increases gene expression.

#2. Regarding my previous question regarding the novelty of BAF155 being responsible for evicting polycomb proteins from chromatin, to which the authors responded "We thank the reviewer for the comments. To the best of our knowledge, the findings reported in the current manuscript for the first time established the antagonism between BAF155 and EZH2."

-In fact, the Crabtree group has published a very detailed study (Nature Genetics volume 49, number 2, february 2017) describing a mechanism whereby BAF proteins evict polycomb complexes including EZH2.

The authors should to cite this paper and discuss how their findings relate to this earlier work.

Point-by-Point Response to The Reviewers Comments

Reviewer #1 (Remarks to the Author):

The authors have adequately addressed the prior concerns. There are no more questions from my end.

Reviewer #2 (Remarks to the Author):

In their revised manuscript, the authors have now completed all of the requested experiments (quite a feat considering the short period of time required for revision). In my opinion, the work is now complete enough, and includes the proper controls, for publication.

A couple of minor textual revisions should be addressed before publication.

#1. In their haste to prepare figures, it seems the Control and knockdown were switched in Supplemental figure 4h. As it stands, it appears Baf155 knockdown increases gene expression.

Response: We thank the reviewer for spotting this. This has now been corrected.

#2. Regarding my previous question regarding the novelty of BAF155 being responsible for evicting polycomb proteins from chromatin, to which the authors responded "We thank the reviewer for the comments. To the best of our knowledge, the findings reported in the current manuscript for the first time established the antagonism between BAF155 and EZH2."

-In fact, the Crabtree group has published a very detailed study (Nature Genetics volume 49, number 2, february 2017) describing a mechanism whereby BAF proteins evict polycomb complexes including EZH2.

The authors should cite this paper and discuss how their findings relate to this earlier work.

Response: We thank the reviewer for the suggestions. This important paper has now been cited (ref. 32) and discussed (first paragraph in the Discussion section).